# Using *apelin*-based synthetic Notch receptors to detect angiogenesis and treat solid tumors

Zhifu Wang[1,5], Fan Wang[1,2,5], Junjie Zhong[1,5], Tongming Zhu[1], Yongtao Zheng[1], Tong Zhao[1], Qiang Xie[1], Fukai Ma [1], Ronggang Li[1,3], Qisheng Tang[1], Feng Xu[1], Xueying Tian[4] & Jianhong Zhu[1✉]

Angiogenesis is a necessary process for solid tumor growth. Cellular markers for endothelial cell proliferation are potential targets for identifying the vasculature of tumors in homeostasis. Here we customize the behaviors of engineered cells to recognize *Apj*, a surface marker of the neovascular endothelium, using synthetic Notch (synNotch) receptors. We designed *apelin*-based synNotch receptors (AsNRs) that can specifically interact with *Apj* and then stimulate synNotch pathways. Cells engineered with AsNRs have the ability to sense the proliferation of endothelial cells (ECs). Designed for different synNotch pathways, engineered cells express different proteins to respond to angiogenic signals; therefore, angiogenesis can be detected by cells engineered with AsNRs. Furthermore, T cells customized with AsNRs can sense the proliferation of vascular endothelial cells. As solid tumors generally require vascular support, AsNRs are potential tools for the detection and therapy of a variety of solid tumors in adults.

[1] Department of Neurosurgery, Fudan University Huashan Hospital, Institute of Brain Science, State Key laboratory of Medical Neurobiology, Shanghai Key Laboratory of Brain Function and Regeneration, Shanghai Medical College, Fudan University, No. 12 Urumqi Mid Road, Shanghai 200040, China. [2] Department of Neurology, Peking University Third Hospital, Beijing, China. [3] Department of Neurosurgery, Shanghai Public Health Clinical Center, Fudan University, No. 2901 Caolanggong Road, Shanghai 201508, China. [4] Key Laboratory of Regenerative Medicine of Ministry of Education, College of Life Science and Technology, Jinan University, Guangzhou, China. [5] These authors contributed equally: Zhifu Wang, Fan Wang, Junjie Zhong. ✉email: jzhu@fudan.edu.cn

A process in the growth and development of humans and other organisms with circulatory systems is angiogenesis, which is attenuated in adults but is robust in solid tumors, which use blood vessels to provide critical support for tumor survival in the form of nutrients and oxygen[1,2]. Specific neovascular markers can be used to identify oncogenesis in adults; therefore, high levels of neovascular markers provide evidence for tumor diagnosis. A sensitive technique for detecting neovascular markers contributes to the early diagnosis of tumors, especially asymptomatic tumors, which is beneficial for cancer treatment. On the other hand, vessel-targeted therapeutic strategies are designed to interfere with the vasculature of tumors to inhibit tumor growth and the strategies developed to inhibit tumor neovascularization have provided therapeutic benefits, including inhibition of vascular endothelial growth factor (*VEGF*) or tyrosine kinases[3]. However, this method is controversial, because recent studies have shown that antiangiogenic therapy may reduce the efficacy of antitumor drugs, and vasculature normalization should not be overlooked[4–6]. Currently, the targets used to inhibit angiogenesis, including *VEGF*, play a fundamental role in both pathological and physiological conditions, thus exposing healthy blood vessels to adverse off-target effects of antiangiogenic therapy. Therefore, there is an urgent need to identify alternative cell surface markers that positively distinguish pathological angiogenesis, such as stable blood vessels from tumor blood vessels, to selectively target pathological angiogenesis.

Currently, in adult murine models, several studies have identified *Apj*, a G protein-coupled receptor, as a potential surface marker of the tumor endothelium. *Apj* is abundantly expressed at the embryonic stage in various tissues, especially in the cardiovascular system, due to the hypoxic microenvironment. In adults, however, high expression of *Apj* is restricted to sprouting vessels[7–9]. The two endogenous ligands of *Apj* are *apelin* and *elabela*, and *apelin* has been shown to be enriched during neovascularization[10]. From the perspective of tumor specificity, the *apelin*/*Apj* system might be a potential target for treatment and diagnosis. However, *apelin* is difficult to detect because of its low concentration in blood; thus, a potential workaround might involve transforming and amplifying the signal of *apelin*/*Apj* for tumor testing, and the cross talk between T cells and endothelial factors offers a means to implement this program. Synthetic Notch (synNotch) receptors have been developed recently to enable the customization of the detection and response behaviors of cells[11,12]. Furthermore, to overcome toxicity in immunotherapy, synNotch receptors are designed to avoid native T-cell responses[13].

To generate synNotch receptors, the transmembrane Notch core domain is retained, whereas the extracellular domain (recognition) and intracellular domain (transcription) can be flexibly modified to match different sensing and response programs[11]. We constructed synNotch receptors based on *apelin* to recognize the surface marker *Apj* on proliferating endothelial cells. After sensing the *Apj*+ endothelial cells, the intracellular domain is released from the plasma membrane and enters the nucleus to stimulate the expression of downstream synNotch pathways (Fig. 1a-d). *Apelin*-based synNotch receptors (AsNRs) support the implementation of the detection and transformation of *apelin*/*Apj* signals, which provides novel evidence for tumor detection.

On the other hand, the successful engineering of immune cells for the clinical treatment of cancer has reaffirmed the native superiority of host immune cells in therapy. Based on immune checkpoint blockades, engineered T cells with chimeric antigen receptors (CARs) have been shown to be impressive prospects for immunotherapy[14,15]. Although CAR T-cell therapies are effective on a set of cancers, the lack of ideal targets and their limited penetration capability prohibit their application to solid tumors. Compared with the vasculature of normal tissues, the tumor vasculature is highly aberrant and dysfunctional. The tumor microenvironment (TME) suppresses the ability of T cells to infiltrate tumors[16,17], and abnormal vessels promote immunosuppressive effects within the TME[18]. The proliferation of solid tumor cells results in a hypoxic environment, which induces the expression of proangiogenic factors, including VEGF and transforming growth factor-β (*TGFβ*). Rapid but sporadic tumor vessel formation impairs the trafficking of effector T cells[1]. In addition, *VEGF* and *TGFβ* promote immunosuppression within solid tumors[19]. Moreover, the complexity of CAR T-cell response programs also contributes to T-cell toxicity, which might lead to cytokine release syndrome[20,21].

In this study, we generated an AsNR to sense sprouting angiogenesis instead of resting endothelial cells (Supplementary Fig. 1a). The *apelin* receptor (*Apj*) is highly expressed in the sprouting endothelial cells but minimally expressed in the cells of other tissues in homeostasis and has been demonstrated as a potential target for tumor therapy[8,9]. We demonstrate that the AsNR can sense proliferating endothelial cells by specifically recognizing *Apj*, and the downstream pathway is mainly controlled by the Tet-Off system. We show that the synNotch receptors are strictly located on the plasma membrane of inactivated cells and that the intracellular domains, i.e., cre recombinases or tetracycline transactivator (tTA) proteins, are cleaved from the receptor and escape into the nucleus after extracellular domains interact with *Apj*. We further customize different synNotch pathways to suit the detection and treatment of solid tumors.

## Results

**Engineered cells with AsNRs can sense *Apj*+ endothelial cells.** To generate a sensor for *Apj*, we selected synNotch receptors developed by Lim and colleagues[12] as a model. The synNotch receptors contain three domains: an extracellular domain that recognizes ligands, a transmembrane core domain that regulates cleavage, and an intracellular domain that stimulates customized programs[11]. The extracellular domain was generated based on *apelin*, which has been identified as an enriched protein in sprouting vessels[7]. Although *apelin* and *elabela* are both endogenous ligands to *Apj*, the investigations of *elabela*-carrying lineages are limited, and our results show that elabela could be activated by the cell line U251/U87 (data not shown). Hypoxia stimulates proliferating endothelial cells to overexpress *Apj*, which is twofold that expressed in normoxia[22]. To improve the specificity of the AsNRs, we modulated the extracellular domain to reduce the sensitivity of *apelin* (Supplementary Table 1) and tested the efficiency of AsNRs in normoxia. With respect to the intracellular domain, to directly test the leakiness of synNotch receptors, we replaced Gal4 (cytoplasmic orthogonal transcription factor) with cre-FLAG (Fig. 1a). After cleavage, tTA diffused into the cytoplasm and nucleus and was difficult to locate; therefore, we used cre recombinases to locate the intracellular domains.

It is critical to examine the leakiness of intracellular domains; therefore, a FLAG-tag was added at the N-terminus of the synNotch receptors for determining the location of cre recombinases (Fig. 1a). The results show that the AsNRs were strictly distributed to the plasma membrane when the engineered cells were not stimulated (Fig. 1b and c). Moreover, the synNotch receptors were able to stably locate to the cytoplasmic membrane after 6-months in cell culture (Supplementary Fig. 1b), and the engineered cells could sense the endothelial cells (Supplementary

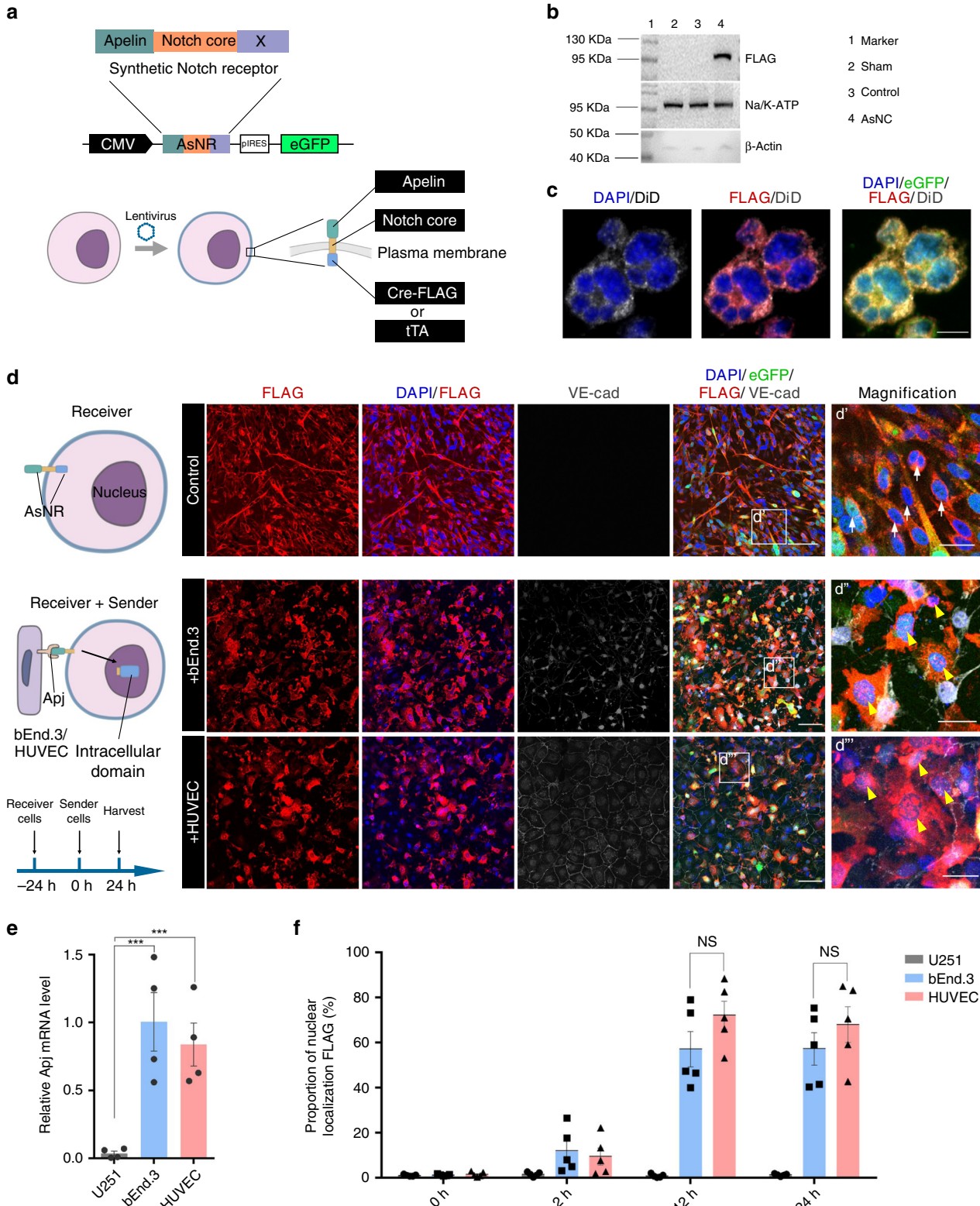

**Fig. 1 Engineered cells with AsNRs can sense *Apj*+ endothelial cells. a** Strategy for generating apelin-based synNotch receptors (AsNRs). A human N-terminal CD8a signal peptide for membrane targeting and intracellular domain (X) contain cre-FLAG and tTA. **b**. Western blot analysis results showing a comparison of FLAG-tag levels on the plasma membrane (intracellular domain: cre-FLAG). **c** Immunostained GFP, FLAG, and DiD (membrane) showing that the intracellular domains were colocalized with the membrane but not the nucleus. **d** Immunostained GFP, FLAG, and VE-cad showing that the intracellular domains entered the nucleus (yellow arrowheads) after the receiver cells contacted the sender cells (bEnd.3 cells and HUVECs). **e** qPCR analysis results showing that *Apj* was highly expressed in both the bEnd.3 cells and HUVECs compared with *Apj* expression in the U251 cells. VE-cad is a marker of endothelial cells ($n = 4$ wells per group). **f** Proportion of intracellular domains entering the nucleus, which showed that cell–cell contact was made within 12 h of sender cell delivery. ($n = 5$ wells per group). Scale bars = 20 μm in **c**, **d′**, **d″**, and **d‴**; 100 μm in **d**. Error bars: SEM. Significance determined by Student's *t*-test: \*\*\**p* < 0.001, n.s. *p* > 0.05.

Fig. 1c), which indicated that these engineered cells could be preserved through passage.

To show that engineered cells (U251 cells) with AsNRs could sense the *Apj*+ endothelial cells, the receiver cells were cocultivated with sender cells, i.e., bEnd.3 mouse brain endothelial cells and human umbilical vein endothelial cells (HUVECs). The intracellular domains were released from the plasma membrane and entered the nucleus after receiver cells contacted the *Apj*+ sender cells (Fig. 1d, e). We quantified the proportion of FLAG-tagged cells localized to the nuclei in the receiver cells between 2 and 24 h after delivering sender cells, which indicated that the receiver cells and the bEnd.3/HUVEC sender cells engaged in cross talk within six hours of cocultivation (Fig. 1f and Supplementary Fig. 2).

These results recapitulated another crucial trait of synNotch receptors: they demonstrated the process by which the intracellular domains enter the nucleus after sensing the target[23,24]. We also demonstrated that the AsNRs have the capability to sense *Apj*+ endothelial cells.

**AsNRs can specifically sense *Apj* and are exclusively induced by proliferating endothelial cells.** To investigate whether the AsNRs exclusively interact with *Apj*, we inhibited the expression of *Apj* in the bEnd.3 cells and HUVECs through transfected RNA interference (RNAi)[22]. *Apj*- bEnd.3 cells and HUVECs were delivered to the receiver cells; however, FLAG-tag staining showed that the intracellular domains did not enter the nuclei (Fig. 2a-d). Moreover, the small proportion of activated receiver cells suggested that AsNR-carrying cells were unable to sense the *Apj*- endothelial cells (Supplementary Fig. 3b). We also delivered cells from different lines (e.g., U87 and astrocytes) to the receiver cells, but the receiver cells were not activated (Supplementary Fig. 3c-f).

Although the interaction between *Apj* and AsNRs was shown, as described above, the capability of nonproliferating endothelial cells to initialize the cleavage of AsNRs was not studied. Thus, we used Ki67-RNAi or colchicine to suppress the proliferation of the bEnd.3 cells and HUVECs (Supplementary Figs. 4 and 5a-e). As expected, the AsNRs could not recognize the nonproliferating bEnd.3 cells or HUVECs (Fig. 2g and Supplementary Fig. 4g), which express low levels of *Apj* (Fig. 2f and Supplementary Fig. 4f), even after cell–cell contact. Subsequently, we constructed an *Apj*+ HEK293 cell line using transfection to generate sender cells that stimulate receiver cells (Supplementary Fig. 5f) and the intracellular domains were detected in the nucleus (Supplementary Fig. 5g). The data e suggested that *Apj* could also interact with AsNRs, regardless of the kinds of cells used as sender cells. Recent studies have demonstrated that *apelin/Apj* is widely expressed in the embryonic and adolescent stages but are dramatically reduced in adults. These results indicated that AsNRs can exclusively recognize proliferating endothelial cells.

**AsNRs can drive customized programs after activation.** As described above, we fused FLAG-tags and cre recombinases to locate the intracellular domains. To test the efficiency of the intracellular domain activation of the customized programs, we generated a new version of AsNRs with tTA intracellular domains[25,26] and the AsNRs, with an Red fluorscent protein (RFP) reporter that was induced by the tetracycline response element (TRE) promoter, were transduced into fibroblasts (Fig. 3a and b). Because of their higher efficiency, the HUVECs were selected as the sender cells. The receiver cells had very low basal activation when sender cells were not added (Fig. 3c), but when we added the sender cells, the receiver cells were activated, and the RFP reporter was highly expressed (Fig. 3d). The results from the flow cytometry analysis showed that the proportion of

activated receiver cells was as high as 80% (Fig. 3e), a finding consistent with previous results.

However, we also wanted to provide evidence that AsNRs work in different types of cells, including various cell lines and primary cells. Thus, HEK293 human embryonic kidney cells and primary mouse astrocytes were utilized to examine the effectiveness of the AsNRs, and the proportion of cells that turned red was similar to that described above (Supplementary Fig. 6b-d).

We also isolated neural progenitor/stem cells (NPSCs) from the spleens of *rosa26-loxp-STOP-loxp-RFP* mice (0–3 days postnatal) and transfected the AsNRs (intracellular domain: cre), which could express the RFP reporter after STOP^flox was deleted by cre recombinases (Supplementary Fig. 6e). Unexpectedly, in our study, the efficiency of the cre recombinases reached ~30%, which was different from the results of previous studies[11]. The most likely reason for this efficacy was that the RFP reporter was knocked-in at the *rosa26* locus and was more sensitive than the GFP reporter[27].

**CD4+ and CD8+ T cells can be customized with AsNRs.** It is critical to customize CD4+ and CD8+ T cells for the purpose of tumor therapy or detection, because T cells can traffic through the circulatory system, and CD4+/CD8+ T cells play a major role in immunotherapy[28,29]. Thus, we tested the properties of AsNRs in T cells by the methods described above (Fig. 4a, b). We sorted the GFP + T cells for the examination (Fig. 4c). These GFP+ T cells turned a deep red after contacting the sender cells (Fig. 4e), which indicated that AsNRs had detected the sender cells and driven the customized program in the T cells. In addition, we analyzed the proportion of activated T cells and the results showed that ~95% of the T cells were activated after the sender cells were added (Fig. 4f, g), a finding that was significantly higher than the previous result (Fig. 3d and Supplementary Fig. 6c, d). There are two potential explanations for this result: (1) adherent T cells made full contact with the HUVECs, because most of the T cells were suspended in the medium, and (2) the size of T cells was much smaller than that of the HUVECs. To support these reasons, we tested the hypothesis that *Apj* is uniformly distributed on the surface of endothelial cells and that fewer endothelial cells lead to dispersed *Apj* protein, resulting in a lower likelihood that the engineered cells would interact with *Apj*. We delivered different concentrations of sender cells to the receiver cells, and 24 hours later, the receiver cells were collected. The results showed that a high density of sender cells activated a high number of receiver cells (Supplementary Fig. 7a-d). We found that, compared with the intensity of mCherry, the RFP reporter is more sensitive and stronger (Supplementary Fig. 7e, f). The receiver cells with the RFP reporter were more likely to turn red than cells with mCherry, a finding consistent with previous reports of lineage tracing studies[30].

**Engineering cells with AsNRs can sense sprouting vessels of tumors in adults.** In contrast to normal adult tissue, solid tumor growth is accompanied by robust sprouting of aberrant vessels that support the metabolism of the tumors[31]. Recent studies indicated that a hypoxic microenvironment is critical to the expression of *Apj*, and that this microenvironment is tightly regulated by *VEGF/VEGFR2*; thus, *Apj* is highly expressed by endothelial cells in solid tumors under TME stimulation[8,32]. The vasculature of tumors shares the substances of the circulatory system and, therefore, engineered cells with AsNRs can be used to detect the endothelium of tumors. The properties of AsNRs necessary to specifically recognize *Apj*+-proliferating endothelial cells in vitro have been shown, but the ability of AsNRs to detect or treat solid tumors in vivo had not yet been investigated. To test

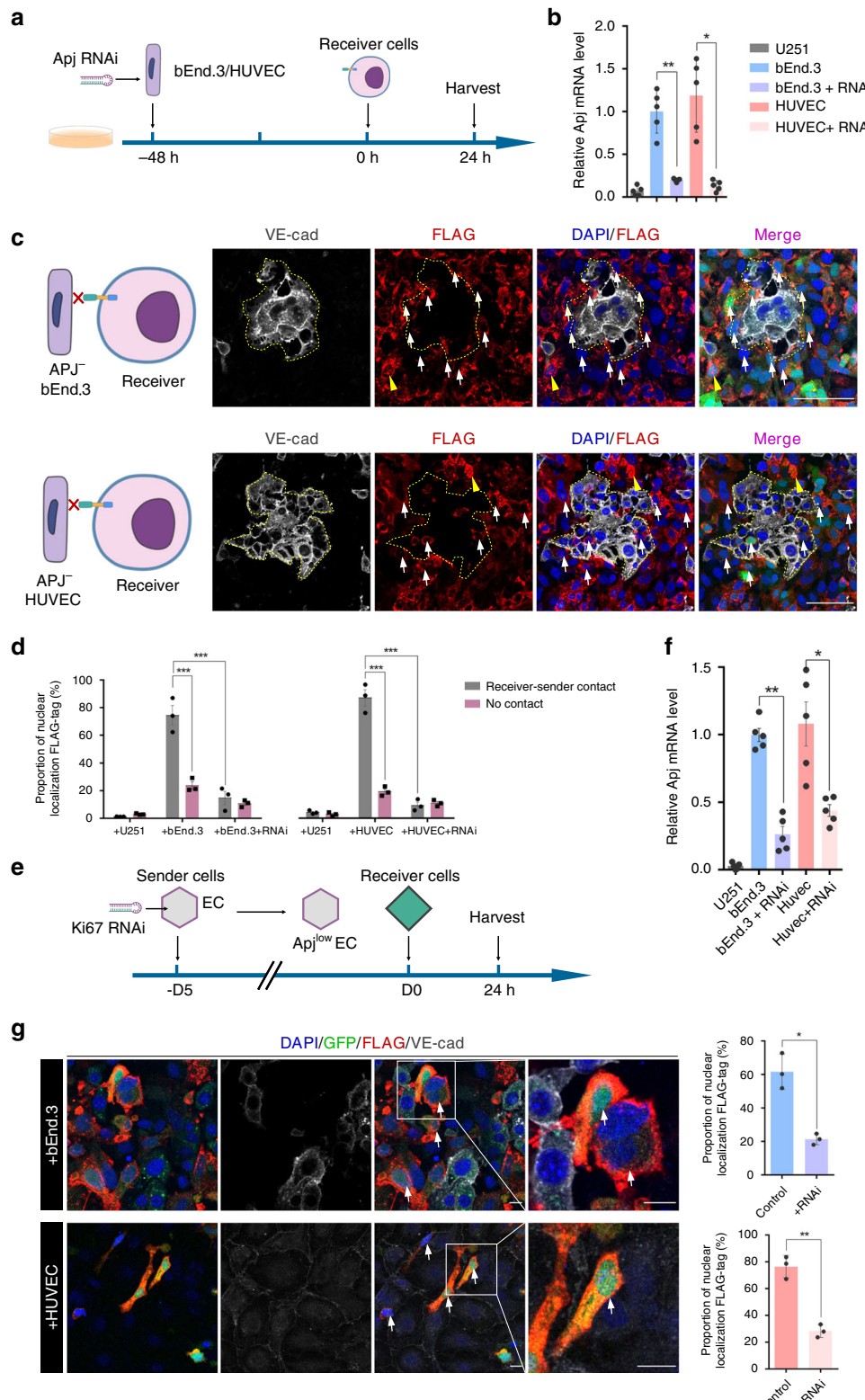

whether AsNRs can function in vivo, we generated tumor models by using the xenograft tumors consisting of Lewis lung carcinoma cell (LLC) and GL261 cells (Fig. 5a)[33]. Then, T cells engineered with AsNRs were intravenously (i.v.) injected and tissues were collected 24 h after injection (Fig. 5a). The engineered cells were present throughout the body, but they were not activated in normal tissues, which have low *Apj* expression (Supplementary Fig. 8). We analyzed the efficiency of engineer cell sensing in vivo,

and ~58% of the engineered cells turned red within 24 hours (Supplementary Fig. 9a, b). The whole-mount fluorescence images showed that the engineered cells turned red in tumors (Fig. 5b), but there were quite few engineered cells with RFP reporters activated in the normal tissues (Fig. 5b), including in the heart, lungs, liver and spleen, corresponding with the *Apj* distribution in adults[7,8,32]. These data indicate that engineered cells with AsNRs could sense the endothelial cells of sprouting vessels

**Fig. 2 AsNRs can specifically sense *Apj* and be exclusively induced by proliferating endothelial cells. a** Experimental strategy to knock down *Apj* in bEnd.3 cells and HUVECs that are cocultured with receiver cells. **b** *Apj* expression in bEnd.3 cells and HUVECs determined by qPCR after RNAi transfection ($n = 5$ samples per group). **c** Immunostained FLAG-tags showing that the intracellular domains were not localized to the nucleus (white arrows) when the *Apj* in sender cells was knocked down. **d** Quantification of nuclear localization FLAG-tags showed that the intracellular domains rarely enter the nucleus without *Apj* ($n = 3$ wells per group). **e** Experimental strategy to inhibit the proliferation of endothelial cells using Ki67-RNAi. See also Supplementary Fig. 4. **f** Expression of *Apj* in bEnd.3 cells and HUVECs determined by qPCR after the proliferation of the bEnd.3 cells and HUVECs was inhibited ($n = 5$ samples per group). **g** The percentage of FLAG-tags localized to nuclei was decreased when the proliferation of sender cells was inhibited compared with the percentage upon receipt of normal senders. Boxed regions are magnified in the right panel ($n = 3$ wells per group). Scale bars = 100 μm in **c**, 50 μm in the magnified panel, 20 μm in **g**. Error bars: SEM. Significance: **$p < 0.01$, *$p < 0.05$.

and that the intracellular domains could activate downstream programs after cell–cell contact. We subsequently attempted to analyze the dynamics and number of RFP$^+$ cells in the blood as the tumors grew; however, because of the low dose of engineered cells and the difficulty in detecting RFP$^+$ cells in real time, we could not distinguish subtle changes in the number of cells. However, in the early stage of tumor formation (7 days post xenograft), we detected the RFP$^+$-engineered T cells in both the blood and tumor tissues (Fig. 5c, d), but it was difficult to detect the RFP$^+$ T cells at the advanced tumor stage (21 days post xenograft; Fig. 5c and Supplementary Fig. 9d-f). In addition, we tested the engineered cells in APC$^{min/+}$ mice, which were used as a spontaneous intestinal adenoma model (Fig. 6a), and the features of the AsNRs were recapitulated in the spontaneous models (Fig. 6b-d). We suggest that, in the capillary lumen of early tumors, the engineered cells may have made sufficient contact with the endothelial cells, and then, a portion of these engineered cells infiltrated the tumor while the others returned to the circulating pool. As the tumors grew, the permeability of the tumor blood vessels to albumin reached tenfold that of normal vessels[34]. Therefore, most of the engineered cells infiltrated the tumor and the few remaining cells were difficult to detect in the blood.

**CD4$^+$ and CD8$^+$ T cells customized with AsNRs targeted and inhibited solid tumors.** To demonstrate that T cells customized with AsNRs are potential tools for cancer treatment, we adjusted the downstream program and replaced the reporter with blinatumomab (Fig. 7a)[13,35]. Blinatumomab is a monoclonal antibody, an α-CD19/CD3 bispecific T-cell engager (BiTE), that directs CD3$^+$ T cells to attack CD19$^+$ tumor cells[36,37]. To avoid inefficient activation, we designed T cells to express AsNRs with the tTa intracellular domain, which stimulated the TRE promoter to express blinatumomab (α-CD19/CD3 BiTE) and the Blue fluorescent protein (BFP) reporter (Fig. 7b and c). The expression of Blinatumomab was represented by the mRNA level, which was determined by quantitative PCR (qPCR) after the engineered cells sensed the proliferating endothelial cells (Fig. 7d). To exert the antitumor function of blinatumomab, we subsequently generated LLC and GL261 cell that produced CD19 before implanting tumors (Supplementary Fig. 10a)[13]. At first, only CD4$^+$/CD8$^+$ T cells engineered with AsNRs were i.v. injected; however, we found that, in the presence of engineered cells alone, the tumor cells were difficult to kill, and the treatment had unsatisfactory results (Fig. 7e and Supplementary Fig. 10d). This outcome was likely due to the lack of free CD3$^+$ T cells. As BiTE combines CD19$^+$ and CD3$^+$ T cells to kill CD19$^+$ tumor cells, that addition of CD3$^+$ cells was expected to enhance the killing effects. We therefore injected CD3$^+$ T cells at the tumor site (Fig. 7b), and the in situ injection reduced the abnormal immune response in the normal tissue and had high specificity. The data showed that the tumor growth was limited and that the tumor volume was significantly reduced during the 25-day period in which the engineered T cells and CD3+ T cells weree injected (Fig. 7e and Supplementary Fig. 10d). Overall, the engineered T cells with

AsNRs targeted tumors and were activated by the endothelial cells of sprouting vessels in vivo, inhibiting tumors by producing α-CD19/CD3 BiTE. It should be noted that *Apj* is expressed during angiogenesis, but not only in tumors. As a result, T cells engineered with AsNRs could be activated by neovascularization, including sprouting collateral vessels in ischemic organs; therefore, they pose a potential risk of cardiovascular toxicity.

**Discussion**

A considerable number of patients diagnosed with cancer have been successfully managed by surgical resection or radiation therapy[38]. However, a portion of lethal cancers (e.g., lung cancer) develop in an asymptomatic form, and by the time they are diagnosed, the disease is already at an advanced stage[39]. In this case, even when the tumors are resected, patients undergo biochemical recurrence, usually because conventional imaging fails to identify regional or distal tumors at the time of diagnosis[40]. Early detection of tumors is therefore a crucial factor in cancer treatment.

Here we develop novel synNotch receptors based on the *apelin/Apj* system, which can customize a variety of cells, including from cell lines and primary cells. Our study demonstrates that engineered cells with AsNRs can be used to specifically target neovascular endothelial cells. In contrast to *VEGF/VEGFR*, *Apj*, which is downstream of *VEGF*, has very limited adult distribution, mainly in endothelial cells during neovascularization, while the complicated *VEGF* family of proteins is widely expressed in various tissues[41,42]. We also provide immunostaining images to demonstrate that the leakiness of these synNotch receptors is acceptable. As T cells can be trafficked throughout the blood vessels of the body, engineered T cells are used to detect the proliferating endothelial cells in animal tumor models; in addition, these T cells have the ability to target tumors. Vascular sprouting is substantially different in tumors that it is in normal adult tissues; hence, the identification of the differences is important for cancer diagnosis. Customized cells with AsNRs provide a new approach for detecting tumors; although subtle signals can currently only be identified after T-cell sorting and it is difficult to further amplify the signal or enhance the sensitivity, we expect that these issues will be resolved in the future.

In recent years, the immunotherapy of tumors has shown a booming trend, especially CAR T, and synNotch receptors are another newly proposed and very promising treatment. synNotch receptors provide a multi-antibody recognition method and has the flexibility to regulate T-cell response programs. Lim and colleagues[12] described a completely different approach to the function of designed immune cells. They showed how synNotch circuits enable researchers to construct new antigen-driven response programs in immune cells that complement and extend endogenous responses to new cells. In particular, a synNotch circuit enables the T-cell sensing and responses to be controlled in a manner that is completely independent of the classical tumor-specific T-cell receptors (TCRs) or costimulatory receptors of the immune system. Therefore, this approach

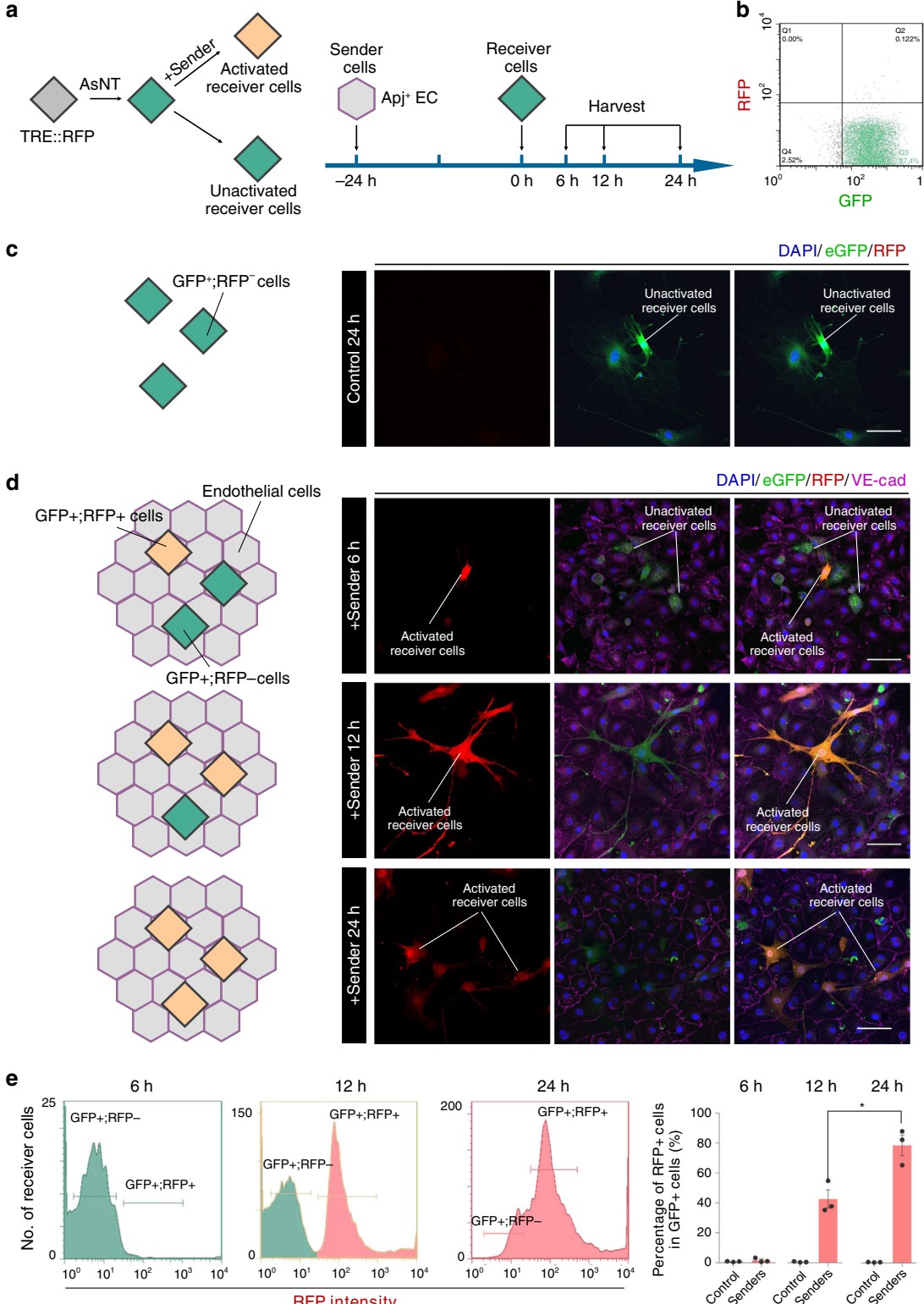

**Fig. 3 AsNRs can drive customized programs after being activated. a** Experimental strategy to test the effectiveness of customized program activation. The RFP reporter gene is promoted by TRE and the reporter is activated after tTA is released from the membrane. Colored diamonds represent receiver cells in different states: green, RFP reporter that is not activated; orange, RFP reporter that is activated. **b** Analysis of receiver cells after sorting by flow cytometry showing that receiver cells did not activate the RFP reporter. **c** Image of fluorescence showed that the RFP reporter in the receiver cells was not activated without contact with the sender cells. **d** After adding HUVECs (sender cells), the receiver cells turned red within 24 h after cell–cell contact. **e** Quantification of the receiver cells showed that the number of RFP + receiver cells gradually increased within 24 h. Scale bars = 50 μm in **c** and **d**. Error bars: SEM. Significance: *$p < 0.05$ ($n = 3$ samples per group in **e**).

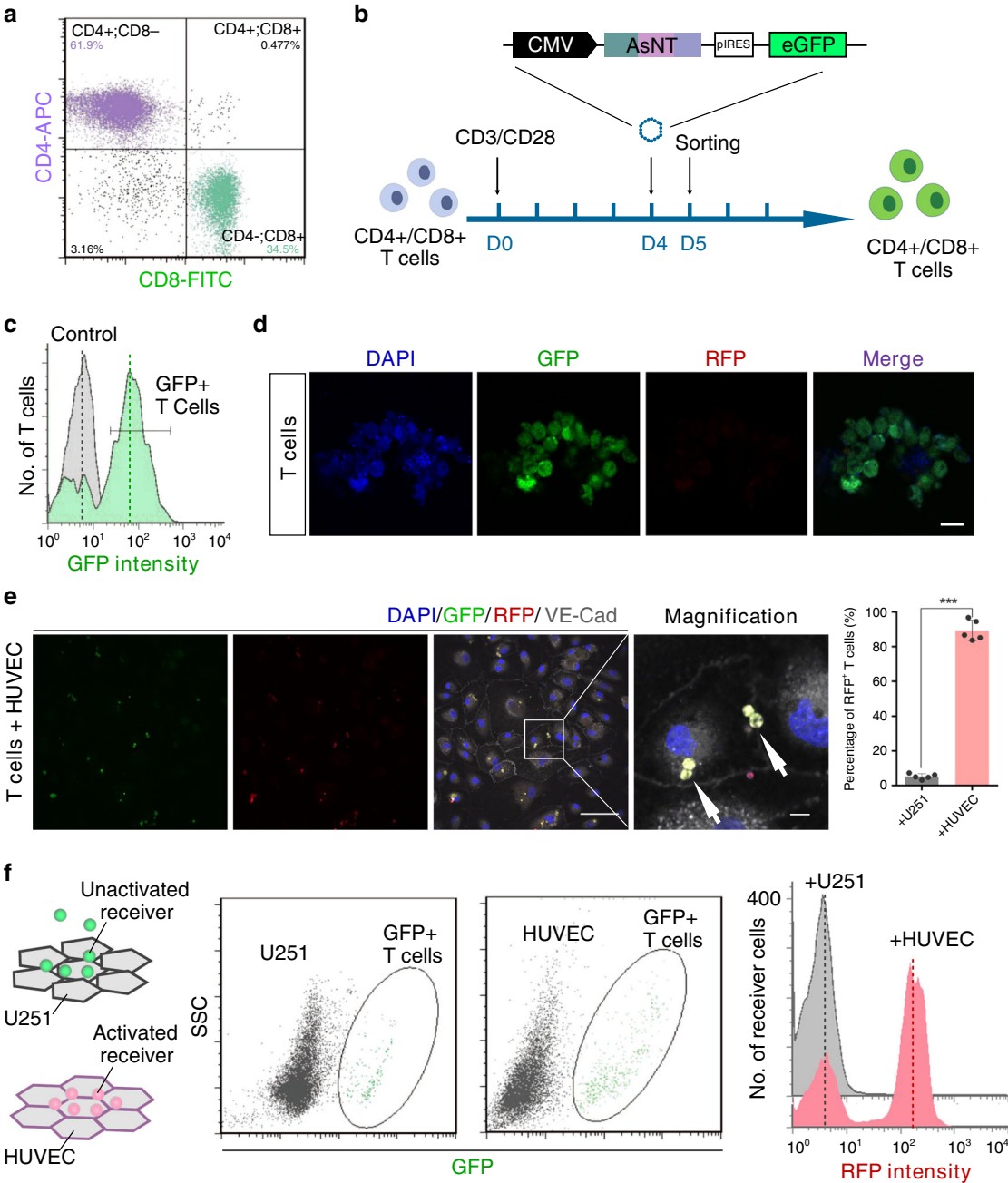

**Fig. 4 CD4+ and CD8+ T cells can be customized with AsNRs. a** In all the experiments, a mixture of CD4+ and CD8+ T cells were used (~2:1). **b**, **c** Due to the low efficiency of T-cell transfection (<20%), GFP-positive T cells were sorted by flow cytometry, and ~85% were retrieved. **d** Image of fluorescence showing very low basal activation in the T cells engineered with AsNRs. **e** Immunostained VE-cad in slices showing that engineered T cells turned red when they engaged in cross talk with endothelial cells (*n* = 5 slices per group). **f** Analysis of adherent T cells showed that HUVECs attracted more engineered T cells than did the U251 cells. The graph on the right shows the number of activated engineered T cells, which indicated that these T cells could sense HUVECs. Scale bars = 10 μm in **d**; 50 μm in **e**, and 10 μm in the magnified panel. Error bars: SEM. Significance: ***p < 0.001.

eliminates the constraints of endogenous systems, allowing for programs with more diverse responses, and in principle, they have a higher degree of controllability and new capabilities. The synNotch receptors have considerable potential to control various types of responses. At present, there have been important advances in several targets associated with tumors, such as ALK, RET, ROS1, and FGFR1/2/3. However, only ALK and ROS1 are approved by the Food and Drug Administration for diagnostic testing, a critical limitation on the further development of immunotherapy for various cancers[43–47].

In this study, the *apelin*-based synNotch circuit we generated can be used not only to detect solid tumors but also to treat solid tumors. AsNRs can effectively recognize the vascular endothelium of tumors in adults and specifically initiate a customized signaling pathway in engineered cells. In contrast to the purpose of previous studies, we did not kill the endothelium to inhibit tumors; we delivered an antibody to tumor cells after T cells recognized sprouting vessels. We demonstrate that AsNRs are effective tools to target and treat tumors. However, the TME, with its high interstitial pressure, resisted the delivery of the drugs, and

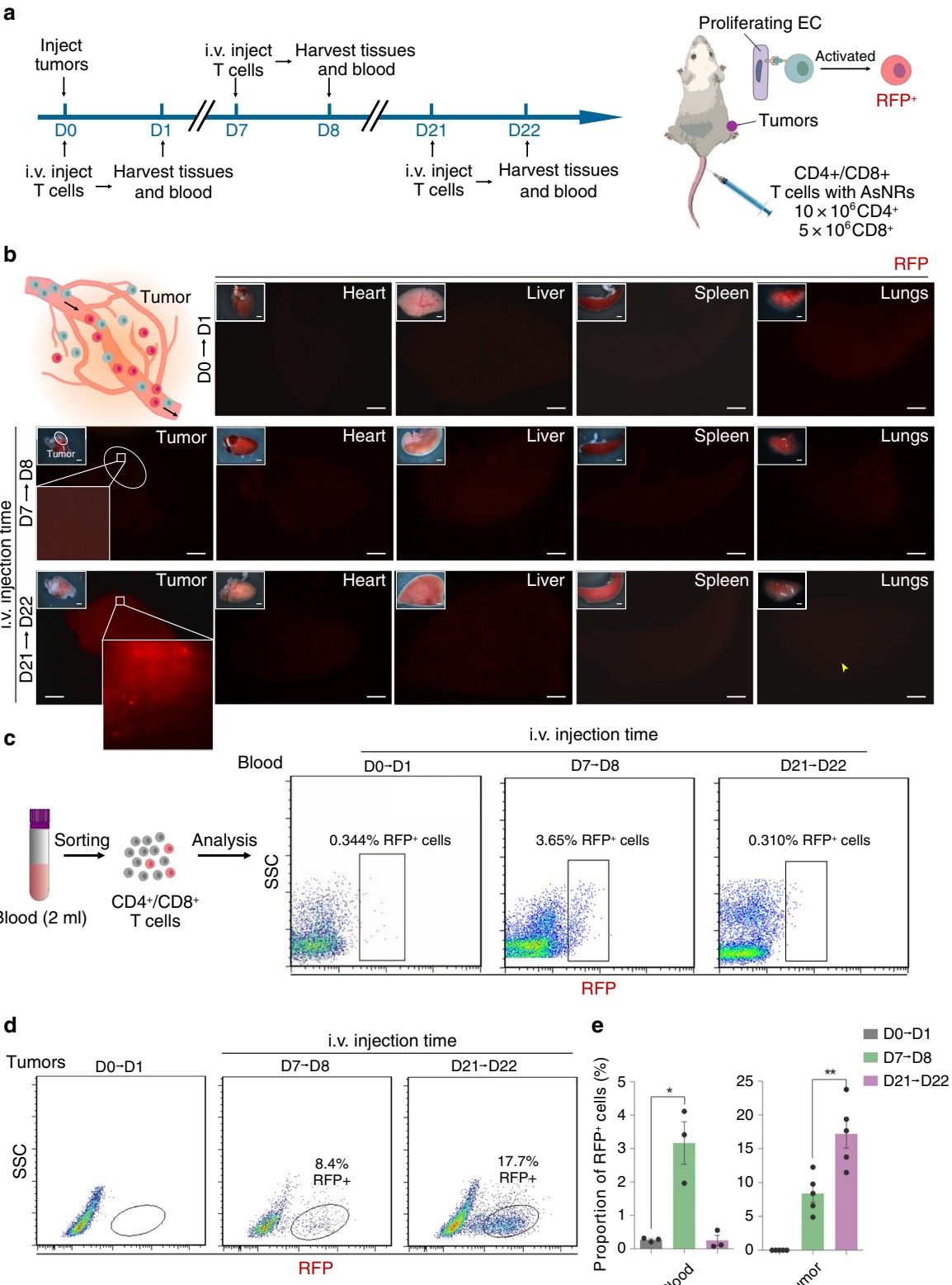

**Fig. 5 Engineered cells with AsNRs can sense the sprouting vessels of tumors in adults. a** Schematic showing the experimental strategy. Tissue and blood were collected at 24 h after engineered cell injection and engineered cells turned red after they sensed sprouting vessels. **b** Whole-mount fluorescence images of tissues showing that engineered T cells failed to turn red in the hearts, livers, spleens, or lungs, but robust RFP signals were detected in tumors, indicating that the AsNRs were specific for *Apj* in vivo. **c** CD4+ and CD8+ T cells from blood were sorted and the proportion of RFP-positive cells increased significantly at D7 compared with those at D0 and D21. **d** The proportion of RFP+ cells increased at D21, indicating that engineered cells were retargeted to tumors as tumors grew. **e** Proportion of RFP+ cells in blood and tumors at D0, D7, and D21. (blood: $n = 3$ samples; tumor: $n = 5$ mice per group). Scale bars = 1 mm in **b**. Error bars: SEM. Significance: *$p < 0.05$.

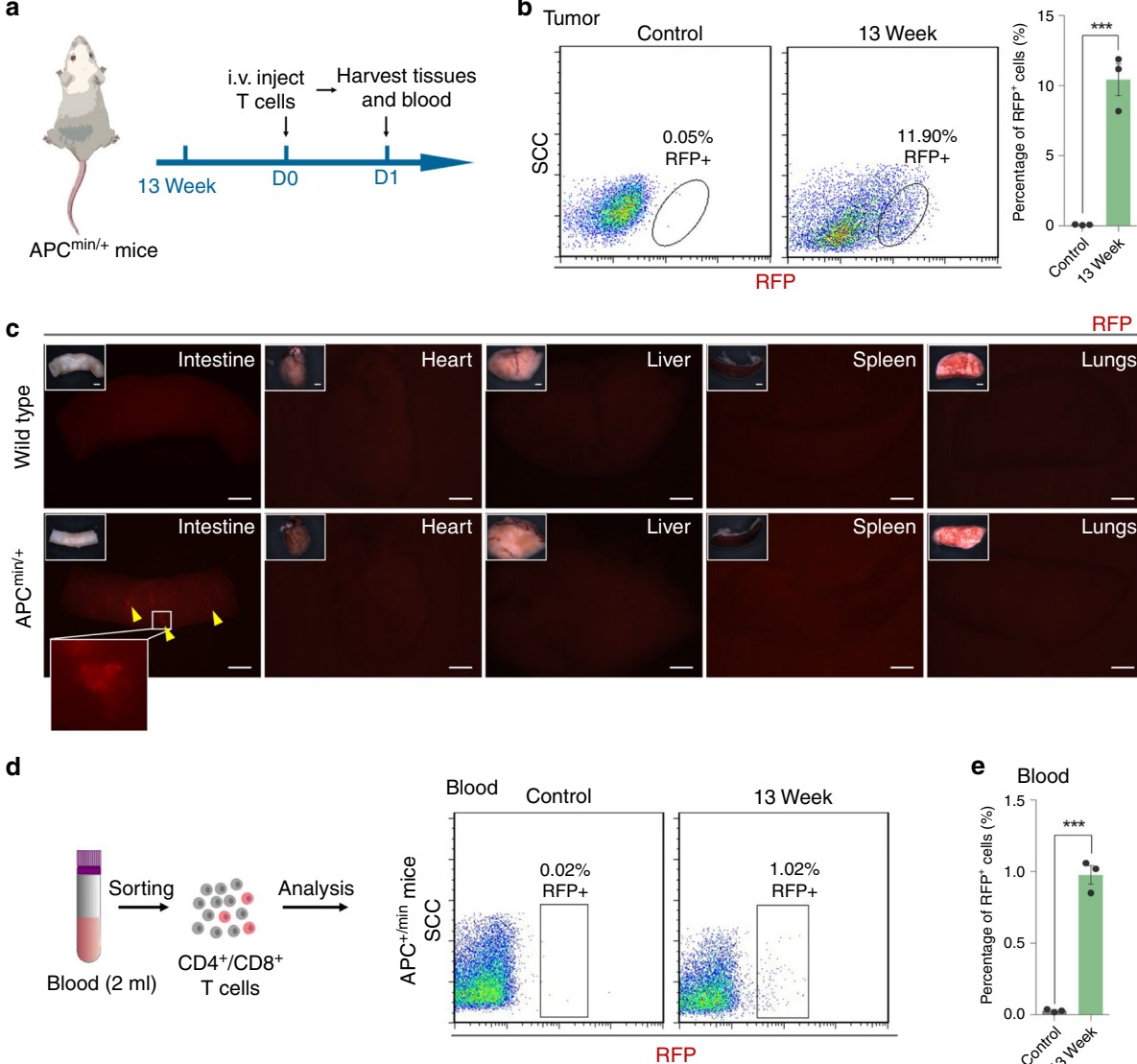

**Fig. 6 Engineered cells with AsNRs can specifically sense the sprouting vessels of spontaneous tumors in adults. a** Schematic showing the experimental strategy using APCmin/+ mice. Engineered T cells were i.v. injected 24 h before the tissues were collected. **b** Quantitative analysis of RFP+ cells in spontaneous tumors determined by FACS and indicating that the engineered cells in the tumors turned red. **c** Whole-mount fluorescence images of tissues, including hearts, lungs, livers, spleens, and intestines, showing that the engineered T cells turned red (yellow arrowheads) in the intestines of APC min/+mice but not in wild-type mice. **d**, **e** Proportion of RFP + cells in the blood of the APCmin/+ mice. Scale bars = 1 mm in **c**. Error bars: SEM. Significance: ***$p < 0.001$, **$p < 0.01$. Each sample consists of blood from two mice ($n = 3$ samples per group).

the tumors could not be blocked completely. Considering that synNotch receptors can be orthogonal to other, different synNotch receptors, AsNRs can be flexibly combined with other synNotch receptors to regulate a signaling pathway, which may enhance the specificity of the solid tumors recognized. Moreover, it is a potential method for normalizing blood vessels while attacking tumor cells through designed AsNR programs.

## Methods

**Model systems and permissions**. All animal procedures were conducted in accordance with the National Institutes of Health Guide for the Care and Use of Laboratory Animals (NIH Publications No. 8023, revised 1978) and approved by the Animal Ethics Committee of Fudan University.

**Animals**. The mice (male, 10–14 weeks old) used in the experiment are all from the C57BL6/J background. *Rosa26-loxp-stop-loxp-RFP* mice were a gift from Zhou Bin's laboratory and all APC<sup>min/+</sup> mice were purchased from GemPharmatech[48–50].

**Cell lines**. HEK293 human embryonic kidney 293 cells (the cell bank of Shanghai Institutes of Biological Sciences), HEK293T (the cell bank of Shanghai Institutes of Biological Sciences), LLC (the cell bank of Shanghai Institutes of Biological Sciences), GL261: murine glioma cell line (Huashan Hospital), U251: U-251 MG (Huashan Hospital), U87: human primary glioblastoma cell line (Huashan Hospital), bEnd.3: mouse brain cell line (the cell bank of Shanghai Institutes of Biological Sciences), HUVECs (the cell bank of Shanghai Institutes of Biological Sciences).

**Apelin-based synNotch receptors design**. AsNRs contain three parts as follows: an extracellular domain, consisting of a human CD8α membrane targeting signal peptide (MALPVTALLLPLALLLHAARP) on the N-terminal[11,13], which is fused to *apelin* (mouse and human share the same amino acid sequence); a transmembrane domain, consisting of the mouse Notch1 core; and an intracellular domain, consisting of tTA or a complex of cre recombinases and FLAG-tags. See also supplemental Table 1.

**Primary T-cell isolation and culture**. CD4+ and CD8+ T cells were isolated from the spleens of mice (C57BL6/J background) by mouse CD4/CD8 (TIL) MicroBeads

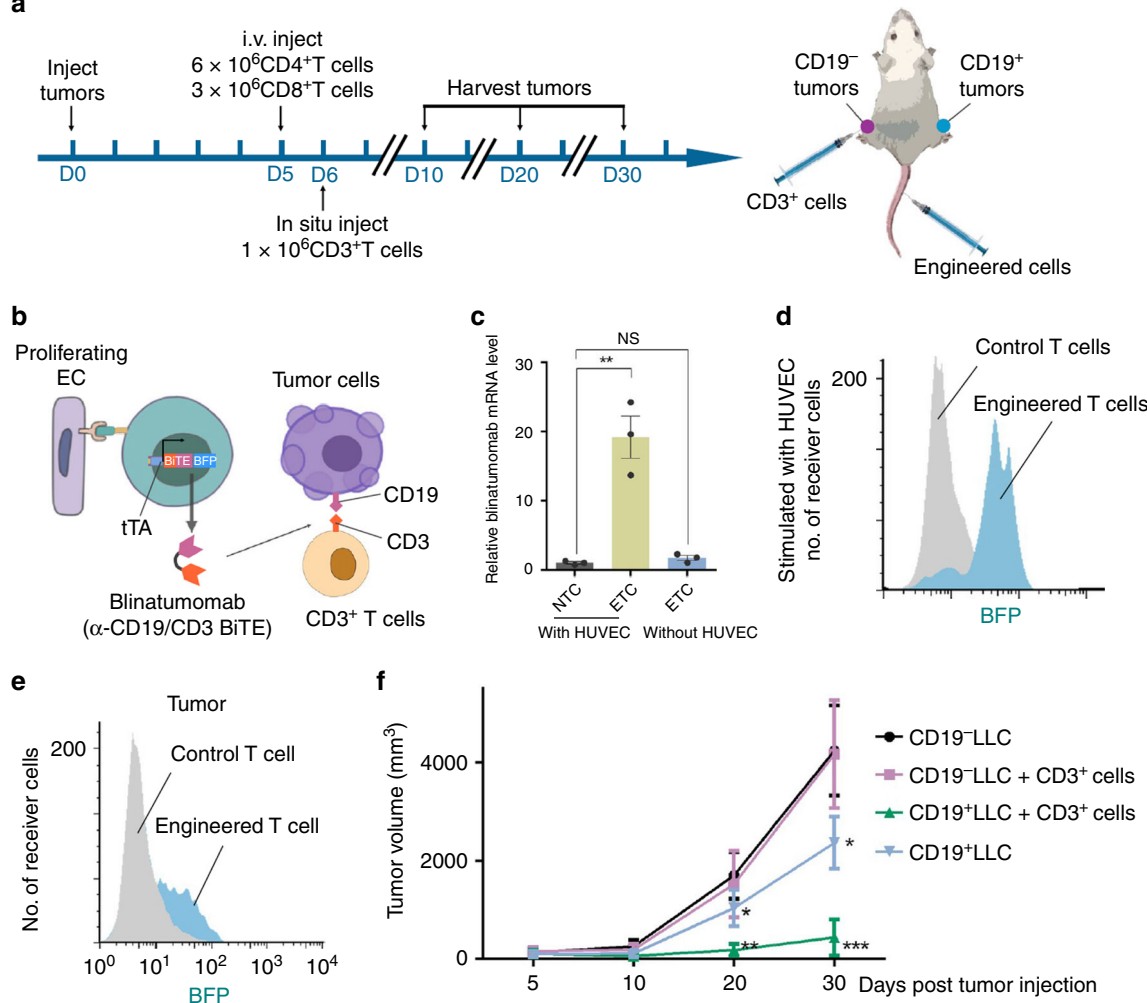

**Fig. 7 CD4+ and CD8+ T cells customized with AsNRs can target and inhibit solid tumors. a** Schematic showing the experimental strategy for **e** and **f**. CD19+ tumor cells were implanted on the right leg and CD19− tumor cells were implanted on the contralateral leg. The engineered T cells were i.v. injected at D5 and CD3+ T cells were injected into tumors. **b** Immunotherapy strategy for treating tumors showing that the engineered T cells secreted BFP reporter and blinatumomab (α-CD19/CD3 BiTE) after they sensed the endothelium of the sprouting vessels, and CD3+ T cells were retargeted to CD19+ tumor cells. **c** Quantitative analysis results showing the mRNA level of blinatumomab when engineered cells were in direct contact with HUVECs. **d** Analysis of the BFP+ cells by flow cytometry after they sensed the HUVECs in vitro. **e** Analysis of the BFP+ cells by flow cytometry showing that the T cells engineered with the BiTE system were activated after sensing tumor endothelium. **f** Tumor growth curves showing that the volume of the tumors was significantly reduced whether or not CD3+ T cells were injected compared with volume of CD19 tumors, but the reduction was more significant after the injection of the CD3+ cells. Error bars: SEM. Significance: ***$p < 0.001$, **$p < 0.01$, *$p < 0.05$ ($n = 3$ in **c–e**, $n = 5$ mice for each group in **f**).

(Miltenyi Biotec, # 130-116-480). T cells were cultivated in RPMI-1640 (Gibco, #11875) with 10% fetal bovine serum (FBS; Gibco, #10437), 10 nM Sodium Pyruvate (Gibco, #11360), 2 mM L-Glutamine, 100 nM non-essential amino acids (Gibco, #11140), and supplemented with 500 U/ml IL-2 (PeproTech, #212125).

**Lentiviral production and transduction.** Gene of interest was amplified by PCR and cloned into lentiviral vector pLenti. Lentivirus was produced via co-transfecting packaging vector(pLP1/2), envelope vector(pLP/VSVG), and transfer vector(pLenti) in 293T cells. Primary CD4+ and CD8+ T cells were stimulated by Dynabeads™ Mouse T-Activator CD3/CD28 (cell : bead, 1 : 3) for 24 h, and lentivirus was then added to the medium for 24 h. Repeat T-cell transductions per 24 h after CD3/CD28 activation. Dynabeads were removed at day 5 post T-cell stimulation. The other cells, including U251, HEK293, primary NSPCs, primary fibroblasts, and primary astrocytes, were all exposed to lentivirus for 24 h. Recombinant DNA used were as follows: pLenti-EF1a-EGFP-P2A-Puro (Obio Technology, ID# HYKY-180322019-DLV), pLenti-EF1a-EGFP-P2A-Puro-CMV-AsNC-3Flag, pLenti-EF1a-EGFP-P2A-Puro-CMV-AsNT, pLenti-EF1a-P2A-Puro- CMV-CD19[13], tetO-RFP-Ubi-puro (Shanghai Genechem, ID# GOSL0186876), Ubi-*Apj*-RNAi-SV40-puro[22], tetO-Blinatumomab-IRES-BFP-SV40-puro (www.drugbank.ca/drugs/DB09052).

**In-vitro stimulation of AsNRs cells.** AsNR cells and bEnd.3 cells/HUVECs were cocultured at a ratio of 1 : 1 in 12-well cell culture plates (for flow cytometry) or 8-well cell culture slides (for fluorescence imaging), and cells were collected after 24 h culture. For inhibition experiments, adding Bevacizumab (HY-P9906;MEC) to the sender cells at 5-day before stimulation.

In Supplementary Fig. 7c, receiver cells were planked at the bottom of an eight-well cell culture slide (BD, #354108) and sender cells were planked on coverslips. Flip coverslips onto the bottom of eight-well cell culture slides to coculture receiver cells and sender cells, and coverslips were removed after 24 h for fluorescence imaging.

**Xenograft tumor models.** LLC and GL261 cells were cultured in DMEM/F12 medium containing 10% FBS, 100 U/ml penicillin, 0.1 mg/ml streptomycin, and glutamine. On day 0, $2 \times 10^6$ CD19− and CD19+ LLC cells were injected into the medullary cavity of the left femur end and the right femur end of the mouse, respectively[33]. For glioma models[51], $1 \times 10^5$ CD19− and CD19+ GL261 cells were injected stereotactically into brain (injection site: bregma, 2 mm to the right of the sagittal suture, 3 mm depth). On day 5 after xenograft tumors, T cells were injected through the tail vein (i.v.). T cells were suspended in phosphate-buffered saline (PBS) for all injections. CD4+ and CD8+ engineered T cells were injected at a ratio of 2 : 1. For the i.v. injection, a total of $9 \times 10^6$ T cells were injected and $5 \times 10^5$ CD3+ T cells were injected into the tumor on the sixth day according to the experimental requirements. In experiments requiring flow cytometry, tumors were collected into RPMI supplemented with 1% FBS (Gibco) on day 7 or day 21 post

tumor injection. Tumors were then chopped with a microscissors and digested for 1 h at 37 °C in RPMI containing 0.1 mg/ml DNase (Worthington # 9003) and 0.2 mg/ml collagenase (Worthington #9001). After incubation, the digested tumors were passed through a 40 μm cell strainer and tumor cells were collected by centrifugation. The cells are then treated with red blood cell lysis buffer and washed with PBS (if the sample is blood, this step is performed directly). Tumors were then stained with LIVE/DEAD Green (Thermo Scientific #34969), and anti-CD4 and anti-CD8 to analyze the expression of tumor infiltrating T cells as well as RFP or BFP. Using Blinatumomab to inhibit tumor growth, tumor growth was measured by calipers for 25 days on day 5, day 10, day 20, and day 30 after tumor implantation.

**CD19 and Blinatumomab design**. CD19 and Blinatumomab were constructed with reference to previous designs[13]. CD19 contains a signal peptide (METDTLLLWVLLLWVPGSTGD) to target the membrane.

**Genomic PCR**. Genomic DNA was extracted from the tails of the mice. Tissues were incubated and lysed in proteinase K overnight at 55 °C and then centrifuged at maximum speed ($21,130 \times g$) for 5 min to obtain supernatants with genomic DNA. The DNA was precipitated with isopropanol, washed in 70% ethanol, and dissolved in deionized water. The oligonucleotides of the primers were listed below[30].

Rosa26-RFP F 5′-GGCATTAAAGCAGCGTATCC-3′
Rosa26-RFP R 5′-CTGTTCCTGTACGGCATGG-3′
Rosa26-RFP WT F 5′-AAGGGAGCTGCAGTGGAGTA-3′
Rosa26-RFP WT R 5′-CCGAAAATCTGTGGGAAGTC-3′

**Immunofluorescence staining**. Mice were perfused with 4% paraformaldehyde (PFA) to collect cells or tissues, which were then washed in PBS to remove excess blood. The brains were then immersed in 4% PFA for 90 min, washed with PBS, and sequentially dehydrated in 20% sucrose and 30% sucrose in PBS. The tissues were embedded in optimal cutting temperature (OCT) compound and cut into 15 μm cryosections on a cryostat. The cryosections or cell culture slides were air-dried for 1 h at room temperature and then blocked with blocking buffer (4% donkey serum and 0.1% Triton X-100 in PBS) for 45 min. Primary antibodies were applied and the sections were incubated overnight at 4 °C, followed by extensive washing with PBS to remove unbound primary antibodies. Colors were developed with secondary antibodies for 1 h at room temperature. Finally, the slides were washed three times in PBS and counterstained with 4′,6-diamidino-2-phenylindole (Vector Labs). Sections with fluorescence reporters were observed and photographed on Nikon (A1+) confocal microscopes (software: NIS-Elements AR 4.60.00; Nikon). Fluorescence images were analyzed by software ImageJ (NIH). Primary and secondary antibodies are listed below.

Rabbit anti-FLAG 1 : 200 Abcam Cat# ab205606
Goat anti-Mouse-GFP (FITC conjugate) Abcam Cat# ab6662
Goat anti-VE-cad 1 : 100 R&D System Cat#AF1002
Rb anti-GFAP 1 : 500 Abcam Cat#ab7260
Chicken anti-GFAP 1 : 800 Abcam Cat#ab4674
Rb anti-RFP 1 : 1000 Rockland Cat#600-401-379
Rat anti-RFP 1 : 200 ChromoTek Cat#ABIN334653
Rb anti-Ki67 1 : 200 Abcam Cat#ab15580
Donkey anti-rabbit IgG H&L (Alexa Fluor® 488) 1 : 1000 Abcam Cat#ab150073
Donkey anti-goat IgG H&L (Alexa Fluor® 488) 1 : 1000 Abcam Cat#ab150129
Goat anti-chicken IgY H&L (Alexa Fluor® 488) 1 : 1000 Abcam Cat#ab150169
Donkey anti-mouse IgG H&L (Alexa Fluor® 488) 1 : 1000 Abcam Cat#ab150105
Donkey anti-rabbit IgG H&L (Alexa Fluor® 555) 1 : 1000 Abcam Cat#ab150074
Donkey anti-rat IgG H&L (Alexa Fluor® 555) 1 : 1000 Abcam Cat#ab150154
Donkey anti-rabbit IgG H&L (Alexa Fluor® 647) 1 : 1000 Abcam Cat#ab150075
Mouse anti-CD3/CD28 Life Technologies Cat#11456D

**Flow cytometry analysis and sorting**. Mice were anesthetized with isoflurane, perfused with 4 °C PBS, and decapitated. Tumors were collected in 37 °C RPMI containing 1% FBS. Small pieces of tumors from mice were treated with 0.2 mg/ml collagenase (1 ml, 37 °C, 15 min) supplemented with 0.1 mg/ml DNase I. Then, 1 ml of prewarmed 10% FBS was added to stop collagenase activity, followed by careful trituration. Blood was collected into anticoagulation tubes (BD) and red blood cells were removed by Red Blood Cell Lysis Buffer. Cell mixtures were passed through a sterile 70 μm filter and resuspended in 4 °C PBS 0.5% bovine serum albumin (BSA) solution for staining. The cells were incubated with primary antibody at 4 °C for 30 min. The fluorescence-activated cell sorting antibodies used are described above. Following washing with 300 μl PBS 0.5% BSA solution, the cells were centrifuged at $300 \times g$ for 3 min and the supernatant was discarded. The secondary antibody was subjected to the same process for staining and washing as above. Finally, 300 μl PBS 0.15% BSA solution was added to resuspend the cells. The data were collected on BD FACSCalibur and BD FACSCanto II (software: BD CellQuest Pro ver5.2) and analyzed by FlowJo (v7.6) software (TreeStar).

Rat anti-CD4-APC 1 : 100 Biolegend Cat#100515
Rat anti-CD8-APC 1 : 100 Biolegend Cat#100712

Rat anti-CD8-FITC 1 : 100 Biolegend Cat#100706
Hamster anti-CD69-APC 1:100 Biolegend Cat#104513
Rat anti-CD3-FITC 1 : 100 Biolegend Cat#100203
APC anti-CD19 1 : 100 Biolegend Cat#115512

**Quantitative RT-PCR**. Tissues or cells were lysed using TRIzol reagent (Invitrogen). RNA was extracted with TRIzol according to the manufacturer's instructions (Invitrogen) and the RNA was converted to cDNA using a Prime Script RT kit (TaKaRa). For the controls, RNA was added to reverse transcription buffer without reverse transcriptase. SYBR Green qPCR master mix (Applied Biosystems) was used and cDNA was amplified on a StepOnePlus real-time PCR system (Applied Biosystems). The sequences for *Apelin* and *Apj* primers were previously described[22].

**Western blotting**. Membrane proteins were isolated by following the protocol of the CelLytic™ MEM Protein Extraction Kit (Sigma, #CE0050)[52]. After centrifugation to separate lysates from the supernatant, proteins were separated using sodium dodecyl sulfate-polyacrylamide gel electrophoresis followed by transfer to a nitrocellulose membrane. Primary antibodies, anti-FLAG and anti-Na-K-ATP, were applied to the nitrocellulose membrane, followed by secondary antibodies conjugated with horseradish peroxidase. Primary and secondary antibodies are listed below.

Mouse anti-FLAG 1 : 5000 Sigma Cat#F1804
Rabbit anti-Na-K-ATP 1 : 100,000 Abcam Cat#ab76020
Goat anti-rabbit IgG H&L (HRP) 1 : 2000 Abcam Cat#ab6721
Goat anti-mouse IgG Beyotime 1 : 3000 Cat#A0216

**Statistical analysis**. The data are presented as the means ± SEM. Cell counting was performed using ImageJ. All data were analyzed with independent-samples *t*-tests (two-sided) using SPSS software (ver. 13.0; SPSS, Inc.) and Prism 6 (graphpad). *$P < 0.05$, **$P < 0.01$, and ***$P < 0.001$ were considered statistically significant, and exact *P*-values were shown in the Supplementary Table 2. All data were determined from three to five independent experiments as indicated in each figure legend. No statistical method was used to predetermine sample size.

**Reporting summary**. Further information on research design is available in the Nature Research Reporting Summary linked to this article.

## Data availability

The source data have been deposited in the Dryad Digital Repository (https://doi.org/10.5061/dryad.9ghx3ffdm)[53]. The data that support the findings of this study are available from the corresponding author upon reasonable request.

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

## Acknowledgements

This work was supported by grants (2018YFA0107900,31771491, 81471242, and 81601069) from the National Nature Science Foundation and Ministry of Science and Technology of China. We thank Professor Sha Hongying of the Institutes of Brain Science of Fudan University for providing technical support for mice construction and also thank Dr Sun Shuhui for technique support of flow cytometry.

## Author contributions

Z.F.W. designed the study. F.W. and Z.F.W. performed these experiments and analyzed the data. J.J.Z performed the flow cytometry. J.H.Z. conceived the study. Q.S.T, Y.T.Z., T.M.Z., F.W. and J.J.Z. bred the mice and performed experiments. Y.T.Z, T.M.Z, Q.X., T.Z., F.X., F.K.M, R.G.L. and J.J.Z provided valuable comments and reagents, and edited the manuscript. X.Y.T. provided valuable comments. Z.F.W. and F.W. analyzed the data and wrote the manuscript.

## Competing interests

The authors declare no competing interests.
