## [Peer Review File · Nature Communications]

Reviewers' comments:

Reviewer #1 (Remarks to the Author):

The authors describe a series of experiments to develop and use a novel synNotch receptor that can be activated specifically by endothelial cells. To achieve this, the authors used the endogenous ligand/receptor pair Apelin/Apj: they generated Apelin-synNotch receptors that can recognize Apj-expressing endothelial cells. They show that these receptors are expressed well in a range of cell lines, and can be activated by endothelial cells in vitro. The authors then moved in-vivo, where they show that T-cells engineered with Apelin-synNotch could be specifically activated in the region of the tumor in a series of different tumor models, both xenograft and spontaneous. Finally, for one of the xenograft model, the authors show that the Apelin-synNotch T-cells can secrete an anti-tumor biologic and thus reduce, at least in part, tumor burden.

Overall the idea is very innovative and exciting. Targeting the tumor vasculature is very promising both for detection and treatment of solid tumors. Using engineered receptors to detect antigens specific for the tumor endothelium is a very interesting idea. The majority of the most important claims of the paper are well supported from the data. One concern with the paper is the difficulty in understanding exactly the experimental setup for each of the experiments shown. Examples see below. In particular Fig. 5. I recommend reviewing text and figures towards increased clarity. Below I write some specific questions or comments to potentially strengthen the manuscript.

SPECIFIC POINTS

Introduction:

On the first read of the introduction, it was not clear why you were talking about detection and diagnosis. A paragraph on stopping vascularization

Rationale: diagnosis and treatment; low in blow, why it matters?

Fig. 1

c) The authors could comment on why some cells with low/no GFP are FLAG-positive; did they note correlation between GFP and FLAG levels (e.g. more GFP—>more FLAG?) If not why?

Fig.2

The RNAi part is very strong; since this is a key result for the rest of the paper, I would suggest to run the analysis by FACS (As they did in Fig.3) as well as the immunofluorescence;

For the part on proliferation, the exclusive use colchicine limits the generality of conclusions that the authors can drive; how can they exclude that the drug is impacting other cellular activities other than proliferation that are relevant for expression and proper trafficking of Apj? There may be another drug that could inhibit proliferation that could provide more control? Or are there ways to obtain "activated" endothelial cells from tumor environments?

Fig. 3

In general, I think reporter activation assays are much more robust than immunofluorescence; I would like to see the key results repeated with FACS analysis, e.g. RNAi, control cells.

e) it is a little unusual in my experience to observe 2 distinct peaks for basal vs induced, usually we see more gradual movement of the whole peak which is consistent with gradual induction of the entire population; any ideas for why you see 2 peaks?

Fig. 5

It took me a while to understand exactly what was going on here. I think spending some time making sure your experimental results are described in a simple and efficacious way would go a long way to increase impact of the paper.

Guiding questions: why are you injecting CD4+ and CD8+ cells at D0, D7 and D21? Are they all engineered cells? For the Day0 cells, when are they harvested?

What happened to the cells injected at Day0, are they still around? Are they still red+? Why did you keep injecting? Are the injected cells always engineered cells?

Overall some of the FACS plots look somewhat strangely skewed, e.g., S6a, why is there such a strong correlation between RFP and GFP? Is it similar to samples from in-vitro studies?

I think one of the strongest result is Supplem Fig. 6g; xenografts could always activate for some other reason, but spontaneous tumors should be very specific; if you have a negative control of another organ I would suggest to put this in the main figures.

One concern that I thought about was whether you can claim that this is specific for Apj, and if you could use an Apj KO mouse to prove the specificity in vivo.

I suspect that the FACS in Fig. 5 are of GFP+ cells, is that correct?

5b) it is very hard to see red cells in non-tumor tissues

Fig. 6

The fact that you have to add CD3+ cells in the location of the tumor pose some limits to the description of this approach as useful to detect tumors (you need to know where the tumor is in order to be able to inject the cells there...). I think you may want to think about defining the scope of the technology in more precise terms. For example, do you think it could work better for smaller tumors?

6c, I found strange to have the relative mRNA level to 1 for the experimental condition, usually it is 1 for control;

“This is likely due to the cell-cell contact between engineered T cells and endothelial cells, which contributes to the lack of free CD3+ T cells.” Do not understand why contact between engineered T cells (CD4+ and CD8+) and endothelial cell contributes to the lack of free CD3+ cells.

6d, experiment not described to enough detail in text/legend.

References, in my version they appear without mention of the journal

Signed:

Leonardo Morsut

Reviewer #2 (Remarks to the Author):

In the manuscript entitled "Using Apelin-based synthetic Notch receptors to detect angiogenesis and treat solid tumors", the authors describe the development of a tumor vasculature-targeting synNotch receptor and the potential uses of this receptor. The article is well-written and the proof-of-concept studies are beautifully designed and executed. The limitations of this work are the lack of immediately translatable relevance (i.e. expression of RFP for detection and secretion of blinatumomab for treatment) and not molecules clinically detectable in serum or bispecific antibodies with relevance to lung carcinoma or glioblastoma. That said, this reviewer understands the proof-of-concept and commends the authors on good work.

Below are minor changes the authors should consider to enhance the work.

- 1) On page 5 of the manuscript, the authors suggest that figure 1b and c demonstrates Apelin-based synNotch receptors distributed on the plasma membrane, however the western blot should contain lanes for quality of membrane fractionation and the immunofluorescent images are not spectacular in terms of distinctively demonstrating membrane localization. The manuscript could benefit from improving those two figures with the suggest of the additional western blot lanes (even as supplemental data) and improved images, potentially with less cells per field.

- 2) In figure 1d, the images chosen for +bEnd.3 and +HUVEC are not ideal and do not match the represented quantification in 1f. The images for HUVEC especially do not demonstrate nuclear localization similar to bEnd.3.

- 3) In figure 1f, the authors demonstrate minimal nuclear localization of FLAG post stimulation at 2 hours and maximal localization at 12 hours. The readers will be very interested in the time between 2 hours and 12 hours. Please repeat these assays with additional times to identify the earliest time of maximal nuclear localization.

4) The data in figure 5b showing activation of RFP in normal tissues of the heart, lungs, liver, and spleen suggests that Apj distribution is not related to sprouting neovascularization and contradicts the therapeutic benefit of this tool. The authors should discuss this normal tissue expression of Apj in the introduction and then clarify in the discussion what this type of expression would mean for treatment options and potential toxicities if AsNRs induce bispecific antibodies or CAR expression in these organs.

5) In the discussion the authors touch on the lack of efficacious targets in solid tumors; however, several promising targets have been identified and some have been utilized clinically while others are making their way to the clinic. It would be beneficial to provide a few sentences describing these targets in the context of AsNRs.

6) In the method for Apelin-based synNotch receptors design, please identify the signal peptide as that of human CD8alpha.

7) In figure 4, the authors describe the requirement to sort GFP+ AsNR cells due to low transduction efficiency, yet the protocol used for lentiviral transduction in 4b is not ideal. The efficiency of lentiviral transduction decreases after 24 hours of T cell activation. Thus, the authors should repeat T cell transductions 24 hours after CD3/CD28 activation and check the transduction efficiency by GFP expression at day 5 - as is noted in the methods section (which is when the authors debeat their cells). Also, the methods are not clear about how the lentivirus is produced. What is the viral envelope for the lentivirus?

Reviewer #3 (Remarks to the Author):

The paper entitled "Using Apelin-based synthetic Notch receptors to detect angiogenesis and treat solid tumors" demonstrates that the AsNRs is an indicator that can be used to probe the interaction between endothelial cells and T cells both using in vitro and in vivo systems to model cancer. The authors do an excellent job in designing and confirming the function of AsNRs on the interaction between endothelial cells and T-cells. The scope of work may be better described in a technical report as the findings describe development of key tools of study but do not uncover novel principles related to the tumor microenvironment. The manuscript is also deficient in a detailed discussion about the specifics of the results and it is felt that stronger evidence is merited to support the use of AsNRs in different settings. The following technical issues should be addressed to improve the quality of this paper.

1. The group synthesized AsNRs with reporter gene Cre-Flag and tTA to investigate AsNRs system and confirm utility in T cells. The synNotch transmembrane core domain included double digestion sites base on Morsut, L. et al. Cell 164, 780–791, 2016. Please provide better details in the AsNRs diagram to understand AsNRs role after cell-cell interaction.
2. Colchicine, an antimetabolic drug which interacts tubulin, and may induce secondary signal to reduce APJ expression. Please check the APJ expression in physiological conditions for responses to colchicine, such as high/low growth factor condition.
3. It is interesting to show AsNRs activation in T cell on U251 and HUVECs in figure 4. Please include an evaluation of APJ expression levels of those cells to clarify that the system working.
4. After administrated the engineered-T cells in mouse tumor model, AsNR activation signal (Red signals) were showed on only tumor which indicates AsNRs were activated in Figure 5. However, the AsNR-T cells will be present throughout the mouse. Therefore, please include AsNRs (GFP signal) to verify AsNR-T cell location in tissues as well as blood. If the engineered T cells are only located on tumors, please explain the reasons and include chemotaxis assay using APJ which might be released on blood steam.
5. It is important to prove APJ level in solid cancer and tissues, as well as blood.
6. Moreover, please show co-localization in between APJ, AsNR-T cells and endothelial cells in tissues.
7. Please include the activation of the modified T cell and normal T cell with blinatumomab system in vitro. Base on Morsut, L. et al., extra- and intra-domain of AsNRs may be digested after ligand interactions which released Apelin from AsNR and the cell junction might be cleavage. The AsNRs cleavage on the engineered T cells may contribute to cell release from APJ positive cells and get T cell potential to kill CD19+ cells.

Dear Editor,

Thank you for your consideration of our manuscript entitled “Using Apelin-based synthetic Notch receptors to detect angiogenesis and treat solid tumors” (Manuscript ID: NCOMMS-19-09267A). We greatly appreciate your comments and those of the reviewers, and we have revised the manuscript accordingly, indicating the changes in red. Our responses to the comments are listed below.

Kind Regards,

Reviewer #1 (Remarks to the Author):

The authors describe a series of experiments to develop and use a novel synNotch receptor that can be activated specifically by endothelial cells. To achieve this, the authors used the endogenous ligand/receptor pair Apelin/Apj: they generated Apelin-synNotch receptors that can recognize Apj-expressing endothelial cells. They show that these receptors are expressed well in a range of cell lines, and can be activated by endothelial cells in vitro. The authors then moved in-vivo, where they show that T-cells engineered with Apelin-synNotch could be specifically activated in the region of the tumor in a series of different tumor models, both xenograft and spontaneous. Finally, for one of the xenograft model, the authors show that the Apelin-synNotch T-cells can secrete an anti-tumor biologic and thus reduce, at least in part, tumor burden.

Overall the idea is very innovative and exciting. Targeting the tumor vasculature is very promising both for detection and treatment of solid tumors. Using engineered receptors to detect antigens specific for the tumor endothelium is a very interesting idea. The majority of the most important claims of the paper are well supported from the data. One concern with the paper is the difficulty in understanding exactly the experimental setup for each of the experiments shown. Examples see below. In particular Fig. 5. I recommend reviewing text and figures towards increased clarity. Below I write some specific questions or comments to potentially strengthen the manuscript.

SPECIFIC POINTS

Introduction:

On the first read of the introduction, it was not clear why you were talking about detection and diagnosis. A paragraph on stopping vascularization

Rationale: diagnosis and treatment; low in blow, why it matters?

Response: Thanks very much for your thoughtful suggestions. According to your suggestion, we discussed the urgent need for diagnosis and treatment in introduction in

the revised manuscript. A highly specific neovascular marker can be used to define oncogenesis in adults, therefore the level of the marker spring up can be used as evidence for tumor diagnosis. A sensitive technique for detecting neovascular marker contributes to diagnose tumors early, especially asymptomatic tumors, which is beneficial for cancer treatment. And targets currently used to inhibit angiogenesis, including VEGF, play a fundamental role in both pathological and physiological conditions, thus exposing healthy blood vessels to adverse off-target effects of anti-angiogenic therapy. Therefore, there is an urgent need to identify alternative cell surface markers that positively distinguish pathological angiogenesis, such as stable blood vessels in tumor blood vessels, to selectively target pathological angiogenesis.

Fig. 1

c) The authors could comment on why some cells with low/no GFP are FLAG-positive; did they note correlation between GFP and FLAG levels (e.g. more GFP—>more FLAG?) If not why?

Response: Thanks for the carefulness of the reviewer. An Internal ribosome entry site (IRES) was inserted between FLAG and GFP, in principle, there is no correlation between GFP and FLAG. We also noted the signal intensity of GFP varies greatly, we attributed this to the difference in copy number caused by lentiviral transfection. And both immunofluorescence and FACS data show that GFP signal intensity is not associated with FLAG levels. Therefore, we think GFP is not linear with FLAG levels. The data are shown below.

Fig.2

The RNAi part is very strong; since this is a key result for the rest of the paper, I would suggest to run the analysis by FACS (As they did in Fig.3) as well as the immunofluorescence;

Response: Thanks for your comment. Following your suggestions, we added the analysis of FACS and immunostaining for accurate quantification at the RNAi part. Our results showed only few receiver cells (~2.5% for FACS, ~6% for immunostaining) were activated, and they are consistent with previous results. These data are shown in the supplementary figure 3.

For the part on proliferation, the exclusive use colchicine limits the generality of conclusions that the authors can drive; how can they exclude that the drug is impacting other cellular activities other than proliferation that are relevant for expression and proper trafficking of Apj? There may be another drug that could inhibit proliferation that could provide more control? Or are there ways to obtain "activated" endothelial cells from tumor environments?

Response: Thank you for your thoughtful comments. As a chemical, colchicine has multiple effects on the cellular activities, and it is hard to exclude these effects. According to your suggestion, we transfected Ki67-RNAi to sender cells in FBS free medium to inhibit their proliferation, as a new control, because we found the Ki67 level of sender cells decreased after adding colchicine. The expression of Apj in HUVEC was significantly down-regulated at 5-day post Ki67-RNAi transfection, and few receivers can be activated by these Ki67-deficient sender cells.

Fig. 3

In general, I think reporter activation assays are much more robust than immunofluorescence; I would like to see the key results repeated with FACS analysis,

e.g. RNAi, control cells.

Response: Thanks for your suggestions. Following your suggestions, we added FACS analysis for RNAi and control cells, indicating that the percentage of receiver cells activated by RNAi and control cells was approximately 2.5% and 0.3%, respectively. The data were shown below, see also supplementary figure 3.

e) it is a little unusual in my experience to observe 2 distinct peaks for basal vs induced, usually we see more gradual movement of the whole peak which is consistent with gradual induction of the entire population; any ideas for why you see 2 peaks?

Response: Thanks very much for the carefulness of the reviewer. We set up multiple experiment control to explain this phenomenon, and then we found that, compared with mCherry, the RFP reporter is more sensitive and stronger. The receiver cells with RFP reporter were susceptible to turn red than cells with mCherry, which is consistent with previous reports of lineage tracing study¹. There was only one peak when the reporter was replaced by mCherry. Therefore, we think the strong intensity of RFP reporter and its sensitivity lead to 2 peaks. Our results are shown below.

Fig. 5

It took me a while to understand exactly what was going on here. I think spending some time making sure your experimental results are described in a simple and efficacious

way would go a long way to increase impact of the paper.

Guiding questions: why are you injecting CD4+ and CD8+ cells at D0, D7 and D21?

Are they all engineered cells? For the Day0 cells, when are they harvested?

What happened to the cells injected at Day0, are they still around? Are they still red+?

Why did you keep injecting? Are the injected cells always engineered cells?

Response: Thanks very much for your comments. We are sorry to make these mistakes.

For each mouse, we injected CD4+ and CD8+ cells (engineered cells) only once, and the mouse was sacrificed the next day to harvest the tissues and blood. These engineered cells are GFP+ cells but not RFP+, while they can turn red after being activated. The mice were injected with engineered cells at D7 or D21 to observe the effect of tumor size on engineered cells, and the D0 was used as a control because there is no tumor formation at D0. The D0 cells were harvested at D1. We have corrected these in the revised manuscript and figures, and the corrected figure is shown below.

Overall some of the FACS plots look somewhat strangely skewed, e.g., S6a, why is there such a strong correlation between RFP and GFP? Is it similar to samples from in-

vitro studies?

Response: Thanks very much for your carefulness. We consulted the technicians of flow cytometry. Since the emission wavelengths of RFP (584nm) and GFP (505nm) used are close to each other, the FACS plots are not ideal despite correcting compensation, but the proportion of RFP+ cells is not significantly affected. To confirm this conclusion, we stained RFP with APC to repeat this experiment. The results are added in the figure S9a of revised manuscript.

I think one of the strongest result is Supplem Fig. 6g; xenografts could always activate for some other reason, but spontaneous tumors should be very specific; if you have a negative control of another organ I would suggest to put this in the main figures.

Response: Thank you for this thoughtful suggestion. Because we showed the treatment data in the last main figure, which is about xenografts, considering the consistency of the manuscript, we put it in the main picture. And we think your suggestion is very important to improve our paper, therefore, we show the distribution of activated engineered T cells in the spontaneous tumor models in figure 6 of the revised manuscript.

One concern that I thought about was weather you can claim that this is specific for Apj, and if you could use an Apj KO mouse to prove the specificity in vivo.

Response: Thank you for your thoughtful comments. We would like to repeat our results with Apj-KO mice, but unfortunately, we fail to get Apj-KO mice in a short time. In our experiments, the RFP reporter can only be expressed after the receiver cells are activated, therefore, we showed the behavior of engineered T cells in different tissues to demonstrate that our tool is specific for Apj. In the adult, Apj expression is restricted

to few tissues, and much lower than sprouting vessels^{2,3}. The whole-mount images showing very few engineered T cells were activated in these tissues, indicating the specificity of AsNT for Apj. In the other hand, we also repeated these results in the tissue sections in the supplementary figure 8 of the revised manuscript.

I suspect that the FACS in Fig. 5 are of GFP+ cells, is that correct?

5b) it is very hard to see red cells in non-tumor tissues.

Response: Thank you for the carefulness. We are sorry again for this inaccurate representation. These engineered T cells are GFP+ cells that, as previously described, can express RFP when they are activated. Therefore, it is hard to see RFP+ cells in non-tumor tissues, which is consistent with the previous results. Because the GFP signal is too weak to identify in whole-mount images, we show it in the tissue sections. See also supplementary figure 8 of the revised manuscript

Fig. 6

The fact that you have to add CD3+ cells in the location of the tumor pose some limits to the description of this approach as useful to detect tumors (you need to know where the tumor is in order to be able to inject the cells there...). I think you may want to think about defining the scope of the technology in more precise terms. For example, do you think it could work better for smaller tumors?

Response: Thank you. That is right. We want to improve the precision of this technology to avoid the cytotoxicity caused by leakiness. But synNotch developed by Morsut, L. et.al was proved as a powerful and flexible tool, which can be combinatorial regulation. Therefore, it can be work without additional CD3+ cells, but we think biosafety and precision is also important. And in our experience, it works better in the early and middle stage.

6c, I found strange to have the relative mRNA level to 1 for the experimental condition, usually it is 1 for control;

Response: Thanks very much for your carefulness. According to your suggestion, we

have corrected it in the revised manuscript.

“This is likely due to the cell-cell contact between engineered T cells and endothelial cells, which contributes to the lack of free CD3+ T cells.” Do not understand why contact between engineered T cells (CD4+ and CD8+) and endothelial cell contributes to the lack of free CD3+ cells.

Response: Thanks for your carefulness. This sentence has been corrected: “This is likely due to the lack of free CD3+ T cells.”

6d, experiment not described to enough detail in text/legend.

Response: Thanks for your suggestions. According to your suggestions. We have corrected it in the revised manuscript.

References, in my version they appear without mention of the journal

Response: Thanks very much for your carefulness. References have been corrected in the revised manuscript.

Signed:

Leonardo Morsut

Reviewer #2 (Remarks to the Author):

In the manuscript entitled "Using Apelin-based synthetic Notch receptors to detect angiogenesis and treat solid tumors", the authors describe the development of a tumor vasculature-targeting synNotch receptor and the potential uses of this receptor. The article is well-written and the proof-of-concept studies are beautifully designed and executed. The limitations of this work are the lack of immediately translatable relevance (i.e. expression of RFP for detection and secretion of blinatumomab for treatment) and

not molecules clinically detectable in serum or bispecific antibodies with relevance to lung carcinoma or glioblastoma. That said, this reviewer understands the proof-of-concept and commends the authors on good work.

Below are minor changes the authors should consider to enhance the work.

1) On page 5 of the manuscript, the authors suggest that figure 1b and c demonstrates Apelin-based synNotch receptors distributed on the plasma membrane, however the western blot should contain lanes for quality of membrane fractionation and the immunofluorescent images are not spectacular in terms of distinctively demonstrating membrane localization. The manuscript could benefit from improving those two figures with the suggest of the additional western blot lanes (even as supplemental data) and improved images, potentially with less cells per field.

Response: Thanks very much for your suggestions. Following your suggestions, we have added new figures for figure 1b and c in the revised manuscript. To demonstrate Apelin-based synNotch receptors distributed on the plasma membrane, we labeled the membrane with DiD probe, indicating the AsNRs are co-localized with membrane. The data are shown in figure 1b and c.

2) In figure 1d, the images chosen for +bEnd.3 and +HUVEC are not ideal and do not match the represented quantification in 1f. The images for HUVEC especially do not demonstrate nuclear localization similar to bEnd.3.

Response: Thanks very much for your carefulness. We have repeated our experiments and improved our images for figure 1f in the revised manuscript. The results are shown in figure 1.

3) In figure 1f, the authors demonstrate minimal nuclear localization of FLAG post stimulation at 2 hours and maximal localization at 12 hours. The readers will be very interested in the time between 2 hours and 12 hours. Please repeat these assays with additional times to identify the earliest time of maximal nuclear localization.

Response: Thank you for this thoughtful suggestion. Following your suggestion, we have added more times to identify the earliest time of maximal nuclear localization. It is interesting that the proportion of nuclear localization increased between 5-hour and 6-hour. The data was shown in supplementary figure 2 of the revised manuscript.

4) The data in figure 5b showing activation of RFP in normal tissues of the heart, lungs, liver, and spleen suggests that Apj distribution is not related to sprouting neovascularization and contradicts the therapeutic benefit of this tool. The authors should discuss this normal tissue expression of Apj in the introduction and then clarify in the discussion what this type of expression would mean for treatment options and potential toxicities if AsNRs induce bispecific antibodies or CAR expression in these organs.

Response: Thanks for your suggestions. We are sorry for unclear representation, in this study, there is quite low proportion of RFP activation in normal tissues, so it does not contradict the therapeutic benefit. It is necessary to discuss the expression profile of Apj in normal tissues. Apj is a G protein-coupled receptor, as a potential surface marker of tumor endothelium. Apj is abundantly expressed at embryonic stage in various tissues, especially in cardiovascular system, due to the hypoxic microenvironment. In the adult, however, the high expression of Apj is restricted to sprouting vessels. We also discussed the potential toxicities in the revised manuscript.

5) In the discussion the authors touch on the lack of efficacious targets in solid tumors; however, several promising targets have been identified and some have been utilized clinically while others are making their way to the clinic. It would be beneficial to provide a few sentences describing these targets in the context of AsNRs.

Response: Thank you for your careful suggestions. Following your suggestions, we discuss current targets for solid tumor in the discussion.

6) In the method for Apelin-based synNotch receptors design, please identify the signal peptide as that of human CD8alpha.

Response: Thanks very much for your careful suggestion. According to your suggestion, we have corrected it in the revised manuscript.

7) In figure 4, the authors describe the requirement to sort GFP+ AsNR cells due to low transduction efficiency, yet the protocol used for lentiviral transduction in 4b is not ideal.

The efficiency of lentiviral transduction decreases after 24 hours of T cell activation. Thus, the authors should repeat T cell transductions 24 hours after CD3/CD28 activation and check the transduction efficiency by GFP expression at day 5 - as is noted in the methods section (which is when the authors de bead their cells). Also, the methods are not clear about how the lentivirus is produced. What is the viral envelope for the lentivirus?

Response: Thank you very much for your suggestions and carefulness. It not only greatly helps our current research, but also supports our future research. We added the new protocol in our methods of the revised manuscript. And the methods of lentivirus production are also revised. Thanks again.

Reviewer #3 (Remarks to the Author):

The paper entitled “Using Apelin-based synthetic Notch receptors to detect angiogenesis and treat solid tumors” demonstrates that the AsNRs is an indicator that can be used to probe the interaction between endothelial cells and T cells both using in vitro and in vivo systems to model cancer. The authors do an excellent job in designing and confirming the function of AsNRs on the interaction between endothelial cells and T-cells. The scope of work may be better described in a technical report as the findings describe development of key tools of study but do not uncover novel principles related to the tumor microenvironment. The manuscript is also deficient in a detailed discussion about the specifics of the results and it is felt that stronger evidence is merited to support the use of AsNRs in different settings. The following technical issues should be addressed to improve the quality of this paper.

1. The group synthesized AsNRs with reporter gene Cre-Flag and tTA to investigate AsNRs system and confirm utility in T cells. The synNotch transmembrane core domain included double digestion sites base on Morsut, L. et al. Cell 164, 780–791, 2016. Please provide better details in the AsNRs diagram to understand AsNRs role after cell-cell interaction.

Response: Thanks very much for your suggestions. Referring to Morsut, L. et al. Cell 164, 780–791, 2016, we discuss more details in the AsNRs diagram in the supplementary figure 1.

2. Colchicine, an antimitotic drug which interacts tubulin, and may induce secondary signal to reduce APJ expression. Please check the APJ expression in physiological conditions for responses to colchicine, such as high/low growth factor condition.

Response: Thanks for your thoughtful suggestions. Following the suggestions, we test the Apj expression in high VEGF and EGF conditions. We find that high VEGF condition can partially rescue the Apj expression in HUVEC. In addition, we demonstrate the effect of long-term inhibition of proliferation on Apj by transfecting Ki67-RNAi in HUVEC. See supplementary figure 4 and 5.

Supplementary figure 4

Supplementary figure 5

3. It is interesting to show AsNRs activation in T cell on U251 and HUVECs in figure. Please include an evaluation of APJ expression levels of those cells to clarify that the system working.

Response: Thank you very much for your suggestions. We have tried many anti-Apj primary antibodies, but none of them work well in the western blot and immunofluorescence. Thus, we test the mRNA levels of U251 and HUVECs to evaluate

the Apj expression levels. The data are shown in the figure 1e.

4. After administrated the engineered-T cells in mouse tumor model, AsNR activation signal (Red signals) were showed on only tumor which indicates AsNRs were activated in Figure 5. However, the AsNR-T cells will be present throughout the mouse. Therefore, please include AsNRs (GFP signal) to verify AsNR-T cell location in tissues as well as blood. If the engineered T cells are only located on tumors, please explain the reasons and include chemotaxis assay using APJ which might be released on blood stream.

Response: Thank you for the thoughtful suggestions. Following your suggestions, we show the AsNR-T cell location in tissue sections, indicating the AsNR-T cells are present throughout the mouse. The numbers of AsNR-T cells are different in different tissues, and very few engineered cells turn red in normal tissues. The data are shown below, or see in the supplementary figure 8.

5. It is important to prove APJ level in solid cancer and tissues, as well as blood.

Response: Thanks very much for your suggestions. According to your suggestions, we test the mRNA levels in solid tumors and tissues using qPCR. In addition, there is very low expression of Apj in normal tissues in adults, as well as blood, as the previous reports²⁻⁷. And the synNotch receptors we designed can only sense robust Apj signals,

which contributes to the specificity for sprouting vessels. We regret that we have not been able to determine protein level of Apj because we have tried seven primary antibodies, but none of them work well. Previous studies have shown that the expression of Apj is restricted to few tissues, especially Zhao et al., 2018, Cell Reports 25, 1241–1254 using lineage tracing to prove the expression profile of Apj in adults, which is consistent with our results.

6. Moreover, please show co-localization in between APJ, AsNR-T cells and endothelial cells in tissues.

Response: Thank you for the suggestions. According to your suggestions, we showed the co-localization between AsNR-T cells and endothelial cells in tissues. Most engineered cells distribute around endothelial cells at 24-hour. The data are shown in supplementary figure 8.

7. Please include the activation of the modified T cell and normal T cell with blinatumomab system in vitro. Base on Morsut, L. et al., extra- and intra-domain of AsNRs may be digested after ligand interactions which released Apelin from AsNR and the cell junction might be cleavage. The AsNRs cleavage on the engineered T cells may contribute to cell release from APJ positive cells and get T cell potential to kill CD19+ cells.

Response: Thanks very much for your thoughtful suggestions. Following your suggestions, we find that blinatumomab system is more sensitive *in vitro*, which is likely due to full cell-cell contact. Also, we find that the engineered T cells *in vivo* will stay away from endothelial cells over time, compared with 24h injection. And this process may be beneficial to kill more CD19+ cells.

References

- 1 Zhang, H. *et al.* Genetic lineage tracing identifies endocardial origin of liver vasculature. *Nature genetics* **48**, 537-543, doi:10.1038/ng.3536 (2016).
- 2 Zhao, H. *et al.* Apj(+) Vessels Drive Tumor Growth and Represent a Tractable Therapeutic Target. *Cell reports* **25**, 1241-1254.e1245, doi:10.1016/j.celrep.2018.10.015 (2018).
- 3 Ho, L. *et al.* ELABELA deficiency promotes preeclampsia and cardiovascular malformations in mice. *Science (New York, N.Y.)* **357**, 707-713, doi:10.1126/science.aam6607 (2017).
- 4 Eyries, M. *et al.* Hypoxia-induced apelin expression regulates endothelial cell proliferation and regenerative angiogenesis. *Circulation research* **103**, 432-440, doi:10.1161/circresaha.108.179333 (2008).
- 5 Liu, Q. *et al.* Genetic targeting of sprouting angiogenesis using ApIn-CreER. *Nature communications* **6**, 6020, doi:10.1038/ncomms7020 (2015).
- 6 Yang, P. *et al.* Elabela/Toddler Is an Endogenous Agonist of the Apelin APJ Receptor in the Adult Cardiovascular System, and Exogenous Administration of the Peptide Compensates for the Downregulation of Its Expression in Pulmonary Arterial Hypertension. *Circulation* **135**, 1160-1173, doi:10.1161/circulationaha.116.023218 (2017).
- 7 Adam, F. *et al.* Apelin: an antithrombotic factor that inhibits platelet function. *Blood* **127**, 908-920, doi:10.1182/blood-2014-05-578781 (2016).

Reviewers' comments:

Reviewer #1 (Remarks to the Author):

I am almost completely satisfied with the revision; there are still a few things that would improve the impact of the paper, and I provide suggestions below; if the authors are able to correct/update in the same way that they improved based on my previous comments the paper should be ready for publication.

Specific good improvements:

fig.2, the addition of another reagent for block of proliferation protects against aspecific effects of colchicine;

Fig.3e, the figure that you put in your rebuttal is great, and I think should be incorporated in the manuscript, e.g. as supplementary;

fig. 5 much easier to follow the experimental setup and description;

fig. 7, explanation of why you ended up injecting CD33 makes much more sense;

Suppl Fig. 8, great to see RFP positive cells close to vessels in the sections!

Things that are still missing in my opinion:

Fig S3

That is a great addition, but the experiments need to be repeated with a positive control, i.e. with a non-relevant RNAi in the sender cells, in the same experiment, to make sure that the receiver cells are not just sick or metabolically, but can be activated;

Suppl Fig.9

Discrepancy with text:

Text says 54% are active at 24h, figures labels D7, which I assume is day 7; where is the data at 24h?

Fig. 5d, 6b, 6d; what does control mean?

Fig. 7c: missing the control of the engineered T cells without HUVEC: important to know how much is the basal expression of the BiTE

Reviewer #2 (Remarks to the Author):

All comments were addressed appropriately.

Reviewer #4 (Remarks to the Author, feedback on authors response to original reviewer#3 concerns):

The revised manuscript adequately addresses most of the reviewer comments. A few additional issues remain:

1. As noted by Reviewer 3, the colchicine experiments are of difficult interpretation, due to the fact that microtubule interactors have multiple cell cycle-unrelated effects and affect a variety of signaling pathways in non-proliferating cells. The point of that experiment is to show that proliferating, but not non-proliferating endothelial cells which presumably express less APJ, are capable of activating AsNR-expressing receiving cells. Supplemental Figure 4, where shRNA to Ki67 was used instead of colchicine, supports that hypothesis and should be shown in the main body of the manuscript, whilst the colchicine experiments should be moved to a supplemental figure.
2. The fact that APJ is not detectable at the protein level is regrettable, but apparently inevitable as multiple primary antibodies have been tried by the authors.
3. The tumor-specificity of AsNR-expressing T-cells is sufficiently well demonstrated in Figures 6 and 7. However, both experiments have a limitation that must be acknowledged. APJ is expressed at all

sites of neovascularization, not just during tumor angiogenesis. As a result, in a real-life translational application, AsNR-carrying T-cells may well be activated at any site of neo-vascularization, including sprouting collateral vessels in ischemic organs in patients with cardiovascular disease. That would pose a potential risk of cardiovascular toxicities. The mouse experiments described in Figures 6 and 7 are not designed to address this potential concern, and this limitation should be clearly acknowledged.

4. Minor point: the English style of the manuscript requires thorough editing for spelling and grammatical errors as well as awkward sentences. For example, the word "receiving" (in reference to receiving cells) is spelled "reciving" or "recieving" in different figures.

Reviewers comments:

Reviewer #1 (Remarks to the Author):

I am almost completely satisfied with the revision; there are still a few things that would improve the impact of the paper, and I provide suggestions below; if the authors are able to correct/update in the same way that they improved based on my previous comments the paper should be ready for publication.

Specific good improvements:

fig.2, the addition of another reagent for block of proliferation protects against aspecific effects of colchicine;

Response: We appreciate your suggestions. Vascular endothelial growth factor (VEGF) is a critical factor for the proliferation of endothelial cells. Bevacizumab(HY-P9906;MEC), a humanized monoclonal antibody, can specifically bind to all VEGF-A isoforms with high affinity, and inhibit its interaction with VEGFR-1 and VEGFR-2. To identify the specific of AsNRs, we used Bevacizumab to inhibit the proliferation of HUVECs, and then we found that the receivers were difficult activated by these HUVECs. Our results are shown below.

Fig.3e, the figure that you put in your rebuttal is great, and I think should be incorporated in the manuscript, e.g. as supplementary;

Response: Thank you very much for your approval. Following your suggestion, we put this figure in Suppl Fig.7e and f.

fig. 5 much easier to follow the experimental setup and description;

Response: Thanks for your carefulness. We corrected the description in the revised manuscript.

fig. 7, explanation of why you ended up injecting CD3 makes much more sense;

Response: We acknowledge our appreciation in this careful suggestion. We added our explanation in the revised manuscript. Blinatumomab (α -CD19/CD3 BiTE) combines CD19+ and CD3+ cells to kill CD19+ tumor cells, and additional CD3+ cells can promote this process.

Suppl Fig. 8, great to see RFP positive cells close to vessels in the sections!

Response: Thanks very much for this thoughtful suggestion. We merged RFP positive cells and vessels in the same panels in Suppl Fig. 8a.

Things that are still missing in my opinion:

Fig S3

That is a great addition, but the experiments need to be repeated with a positive control, i.e. with a non-relevant RNAi in the sender cells, in the same experiment, to make sure that the receiver cells are not just sick or metabolically, but can be activated;

Response: Thanks very much for your thoughtful comment. Following this suggestion. We added the group of non-relevant RNAi in Suppl Fig. 3. The results are shown below.

Suppl Fig.9

Discrepancy with text:

Text says 54% are active at 24h, figures labels D7, which I assume is day 7; where is the data at 24h?

Response: We apologize for this imprecise mark. In the revised manuscript, D7 was changed to D7-D8 in Suppl Fig.9.

Fig. 5d, 6b, 6d; what does control mean?

Response: Thank you for this comment. In these figures, control means the first day post tumor injection. We replaced 'control' with D0-D1 in the revised manuscript.

Fig. 7c: missing the control of the engineered T cells without HUVEC: important to know how much is the basal expression of the BiTE

Response: Thank you very much for this suggestion, it is important to show the baseline

of the BiTE. We tested the expression of the BiTE in the engineered T cells, and there is no leakiness when they did not contact with HUVEC. The data is shown below, see also figure 7c.

Reviewer #2 (Remarks to the Author):

All comments were addressed appropriately.

Reviewer #4 (Remarks to the Author, feedback on authors response to original reviewer#3 concerns):

The revised manuscript adequately addresses most of the reviewer comments. A few additional issues remain:

1. As noted by Reviewer 3, the colchicine experiments are of difficult interpretation, due to the fact that microtubule interactors have multiple cell cycle-unrelated effects and affect a variety of signaling pathways in non-proliferating cells. The point of that experiment is to show that proliferating, but not non-proliferating endothelial cells which presumably express less APJ, are capable of activating AsNR-expressing receiving cells. Supplemental Figure 4, where shRNA to Ki67 was used instead of colchicine, supports that hypothesis and should be shown in the main body of the manuscript, whilst the colchicine experiments should be moved to a supplemental figure.

Response: We appreciate your thoughtful suggestions. As you remark, the colchicine experiments are of difficult interpretation, thus we replace the colchicine parts with the Ki67-RNAi experiments in the main figures. Furthermore, we added Bevacizumab to inhibit the proliferation of endothelial cells to support our findings. Our data are shown below.

2. The fact that APJ is not detectable at the protein level is regrettable, but apparently inevitable as multiple primary antibodies have been tried by the authors.

Response: Thank you very much for your understanding. We will try to investigate the properties in the future.

3. The tumor-specificity of AsNR-expressing T-cells is sufficiently well demonstrated in Figures 6 and 7. However, both experiments have a limitation that must be acknowledged. APJ is expressed at all sites of neovascularization, not just during tumor angiogenesis. As a result, in a real-life translational application, AsNR-carrying T-cells may well be activated at any site of neo-vascularization, including sprouting collateral vessels in ischemic organs in patients with cardiovascular disease. That would pose a potential risk of cardiovascular toxicities. The mouse experiments described in Figures 6 and 7 are not designed to address this potential concern, and this limitation should be clearly acknowledged.

Response: Thank you very much for this thoughtful suggestion. Patients, especially those with cardiovascular disease, may have vascular hyperplasia, which increases the toxicity of T cells. It is a critical limitation for AsNR, but almost all vascular markers have this problem, including VEGF. In contrast, AsNR is more specific for vascular targeting. We discussed the potential risks of AsNR for patients with cardiovascular disease in the revised manuscript. And we will develop more tools to improve the specific of our engineered cells in the future.

4. Minor point: the English style of the manuscript requires thorough editing for spelling and grammatical errors as well as awkward sentences. For example, the word "receiving" (in reference to receiving cells) is spelled "reciving" or "receiving" in different figures.

Response: Thank you very much for your careful suggestions. We apologize for our mistakes. Following your suggestions, we improved the English writing in the revised manuscript.

REVIEWERS' COMMENTS:

Reviewer #1 (Remarks to the Author):

All my comments regarding the scientific merit have been addressed;

there is only one missing panel that I think is a kind of typo:

in my pdf file Fig. 7c is missing; in figure 7, there are 7b and then 7d without Fig. 7c, which was a good addition presented in rebuttal, that I think should be incorporated in the final version of the manuscript;

I am still a little concerned about the style, grammar and syntax, which still sounds very odd and hard to navigate at first sight, and even for me that I have seen it multiple times at this point. I wish I had time to help with this, but I don't right now. I hope you can find some editorial help from some other source? Your paper deserves it!

Leonardo Morsut

REVIEWERS' COMMENTS:

Reviewer #1 (Remarks to the Author):

All my comments regarding the scientific merit have been addressed; there is only one missing panel that I think is a kind of typo: in my pdf file Fig. 7c is missing; in figure 7, there are 7b and then 7d without Fig. 7c, which was a good addition presented in rebuttal, that I think should be incorporated in the final version of the manuscript;

I am still a little concerned about the style, grammar and syntax, which still sounds very odd and hard to navigate at first sight, and even for me that I have seen it multiple times at this point. I wish I had time to help with this, but I don't right now. I hope you can find some editorial help from some other source? Your paper deserves it!

Leonardo Morsut

Response: We appreciate your suggestions. We corrected it in the revised manuscript, and invited an expert in AJE (American Journal Experts) to polish the English style.